# Dual-barcoded shotgun expression library sequencing for high-throughput characterization of functional traits in bacteria

Vivek K. Mutalik [1], Pavel S. Novichkov[1], Morgan N. Price[1], Trenton K. Owens[1], Mark Callaghan[1], Sean Carim [2], Adam M. Deutschbauer[1,2] & Adam P. Arkin [1,3]

A major challenge in genomics is the knowledge gap between sequence and its encoded function. Gain-of-function methods based on gene overexpression are attractive avenues for phenotype-based functional screens, but are not easily applied in high-throughput across many experimental conditions. Here, we present Dual Barcoded Shotgun Expression Library Sequencing (Dub-seq), a method that uses random DNA barcodes to greatly increase experimental throughput. As a demonstration of this approach, we construct a Dub-seq library with *Escherichia coli* genomic DNA, performed 155 genome-wide fitness assays in 52 experimental conditions, and identified overexpression phenotypes for 813 genes. We show that Dub-seq data is reproducible, accurately recapitulates known biology, and identifies hundreds of novel gain-of-function phenotypes for *E. coli* genes, a subset of which we verified with assays of individual strains. Dub-seq provides complementary information to loss-of-function approaches and will facilitate rapid and systematic functional characterization of microbial genomes.

[1] Environmental Genomics and Systems Biology Division, Lawrence Berkeley National Laboratory, Berkeley, 94720, CA, USA. [2] Department of Plant and Microbial Biology, University of California, Berkeley, 94720, CA, USA. [3] Department of Bioengineering, University of California, Berkeley, 94720, CA, USA. Correspondence and requests for materials should be addressed to V.K.M. (email: vkmutalik@lbl.gov) or to A.P.A. (email: aparkin@lbl.gov)

Advances in DNA sequencing have had a tremendous impact on microbial genomics, as tens of thousands of genomes have now been sequenced[1]. However, only a small fraction of these microorganisms have been experimentally studied and as such, our predictions of gene function, metabolic capability, and community function for these microorganisms are based largely on automated computational approaches[2]. Unfortunately, many of these computational predictions are incomplete or erroneous, especially in instances where the homology of a sequenced gene is too distant from any experimentally characterized relative[3]. To bridge this gap between sequencing and functional characterization, it is imperative that large-scale, inexpensive, and organism-agnostic tools are developed and applied[4].

A number of large-scale approaches based on loss-of-function genetics have been developed for microorganisms including gene-knockout libraries[5–9], recombineering based methods[10,11], transposon mutagenesis coupled to next-generation sequencing (TnSeq)[12,13], and CRISPR interference (CRISPRi)[14]. Collectively, these strategies all rely on measuring the phenotypic consequences of removing a gene from a microorganism and inferring protein function based on these phenotypes. An adaptation of TnSeq that incorporates and uses random DNA barcodes (RB-TnSeq) to measure strain abundance in a competitive growth assay[13] has recently been applied on a larger scale to identify mutant phenotypes for thousands of genes across 32 bacteria[15]. Despite their utility, these loss-of-function approaches suffer some limitations: only CRISPRi is effective for interrogating essential genes under multiple conditions, it is challenging to identify phenotypes for genes with redundant functions using single mutants, and these approaches require some degree of genetic tractability in the target microorganism.

A complimentary approach for studying gene and organism function is to generate gain-of-function overexpression libraries and analyze the phenotypic consequences of increased gene dosage. Indeed, the impact of enhanced gene dosage on adaptation and evolution are well documented across all three kingdoms of life and have been shown to be an important contributor to numerous diseases and drug resistance phenotypes[16–18]. Increased gene copy or overexpression as a genetic tool has a rich history of connecting genes to cellular functions and has been exploited as a versatile screening technique to identify drug targets[16,19,20], antibiotic and metal resistance genes[17,21,22], virus resistance genes[23], genetic suppressors[24,25], as well as for a number of chemical genomics[8,9] and biotechnology applications[26–28]. Although a number of technologies have been developed for overexpression screens including defined open reading frame (ORF) libraries[6,20,29] and activation modes of recombineering[30,31], transposon insertions[32], or CRISPR systems[33], these strategies are limited, either due to the need for expensive and laborious generation of archived strains or the need for organism-specific genetic tools.

A simpler alternative for overexpression screens is a shotgun library-based approach in which random DNA is introduced into a host organism for phenotyping and functional assessment. This approach has been widely used for studying increased copy number effects on a desired phenotype[26,27] and for activity-based screening of metagenomic samples[34,35]. Nevertheless, most shotgun expression libraries have only been assayed in a small number of conditions looking for a specific gene function, and are often performed as qualitative selections on a plate[34–36]. Furthermore, current shotgun-based approaches typically require tedious and expensive sequencing and sample preparation protocols for identifying the selected gene(s)[26,27,37,38]. With arrival of next-generation sequencing technologies, all positive candidates can be pooled, and cloned regions can be amplified and sequenced in parallel[39,40]. Unfortunately, sequencing the cloned regions (to identify the genes conferring the phenotype) is labor intensive and may become cost-prohibitive if the overexpression library is being assayed under many conditions. As such, there is a need for high-throughput gain-of-function technology that is simple, quantitative, agnostic to source DNA, and which facilitates multiplexed quantification of fitness under hundreds of experimental conditions. Here we present a new method termed Dub-seq, or dual-barcoded shotgun expression library sequencing, for performing high-throughput and quantitative gain-of-function screens. Dub-seq requires an initial characterization of the overexpression library by linking the genomic breakpoints of each clone to a pair of random DNA barcodes. Subsequent screens are performed using a competitive fitness assay with a simple DNA barcode sequencing and quantification assay (BarSeq[41]). As a demonstration of this approach, we generate an *Escherichia coli* (*E. coli*) Dub-seq library and assayed the phenotypic consequences of overexpressing nearly all genes on *E. coli* fitness under dozens of experimental conditions. We show that Dub-seq yields gene fitness data that is consistent with known biology and also provides novel gene function insights. We validate some of these new findings by overexpressing individual genes and quantifying these strains' fitness. Given that only DNA and a suitable host organism for assaying fitness are necessary and not the genetic tractability of the organisms of interest, Dub-seq can be readily extended to diverse functional genomics and biotechnology applications including functional interrogation of DNA from uncultivated clades.

## Results

**Overview of Dub-seq.** The Dub-seq approach involves cloning a shotgun expression library between two random DNA barcodes and associating the precise breakpoints of the DNA fragments to the barcodes prior to measuring phenotypes. To assess the fitness of individual strains carrying these plasmids, DNA barcode sequencing (BarSeq)[41] is then employed, which is simple and amenable to large-scale sample multiplexing. The Dub-seq approach is summarized in Fig. 1 and can be separated into four different steps. First, a plasmid library is generated with pairs of random 20 nucleotide DNA sequences, termed the UP and DOWN barcodes. To link the identities of the two-barcode sequences on each plasmid, Barcode Pair sequencing (BPseq) is performed (Fig. 1a, Methods). Second, sheared genomic DNA from an organism under investigation is cloned between the previously associated UP and DOWN barcodes (Fig. 1b). Third, the genomic fragment endpoints are mapped and associated with the two-barcode sequences using a TnSeq-like protocol[13]. We term this step Barcode-Association-with Genome fragment by sequencing or BAGseq and the resulting plasmid library as the "Dub-seq" library (Fig. 1c). The BAGseq step requires two sample preparations to separately map genomic fragment junctions to the UP and DOWN barcodes. The BAGseq characterization generates a table of barcode sequences and the cloned chromosomal breakpoints at single-nucleotide resolution. Because the two random DNA barcodes have been previously associated, we can infer the exact sequence of each plasmid in the Dub-seq library if the sequence of the source DNA is known. Finally, we introduce the Dub-seq plasmid library into a host bacterium and monitor the fitness of strains carrying these plasmids in a competitive fitness assay under a particular condition by PCR amplifying and quantifying the abundance of the DNA barcode sequences (BarSeq[41], Fig. 1d). In these pooled fitness experiments, the barcode abundance changes depending upon the fitness phenotype imparted by the barcode-associated genome fragments. A data analysis pipeline yields fitness scores for individual strains (or "fragments") and for each gene. These gene scores provide an

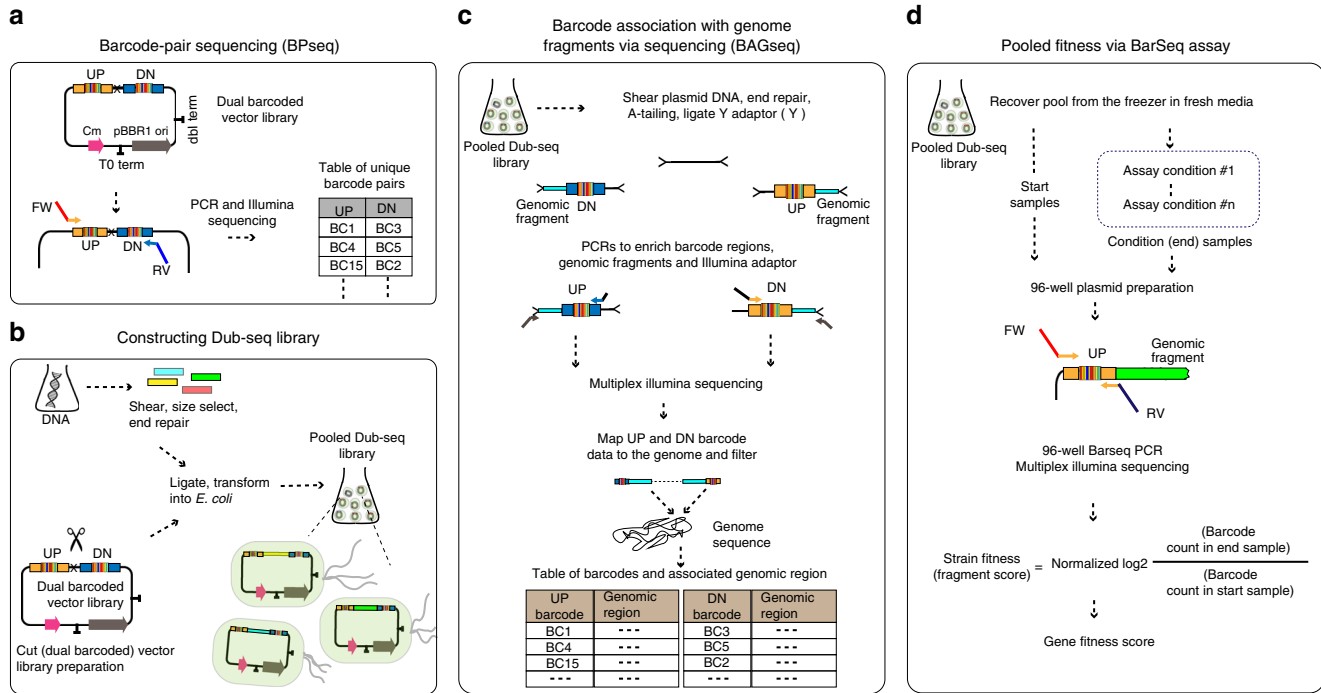

**Fig. 1** Schematic overview of the dual-barcoded shotgun expression library sequencing (Dub-seq) approach. **a** A pair of random 20 nucleotide DNA sequences, the UP and DOWN (DN) barcodes are cloned into an expression vector. Deep sequencing of the dual-barcoded vector (BPseq) associates UP and DOWN barcode sequences. **b** Target genomic DNA is randomly sheared and cloned between the UP and DOWN barcodes to create the Dub-seq plasmid library. **c** To characterize the Dub-seq library, a "Tn-seq" like protocol is performed to precisely map the two genomic breakpoints of each insert and to associate each breakpoint with its random DNA barcode sequence. If the source genome(s) has been sequenced, then BAGseq can be used to define the exact sequence of each plasmid in the library. **d** The fitness of bacteria carrying different plasmids can be measured with pooled growth assays and deep sequencing of the DNA barcodes (BarSeq). Strain (or fragment) fitness is defined as the $\log_2$ ratio of barcode abundance after selection (end) versus before (start). Gene fitness is estimated from the fragments' fitness by a constrained regression

assessment of the phenotypic consequence of overexpressing nearly all of the genes represented in the cloned DNA fragments. The advantage of Dub-seq is that it decouples the characterization of a shotgun overexpression library (which is more laborious) from the cheaper and simpler fitness determination step using BarSeq. As such, a Dub-seq library can be readily assayed in hundreds of different experimental conditions. Dub-seq can be viewed as an overexpression-based, gain-of-function version of our previously described method for random barcode transposon site sequencing (RB-TnSeq)[13].

**Generation of *E. coli* Dub-seq library**. To generate a Dub-seq library, we used a broad host range medium copy vector with a modified pBBR1 replication origin. We used standard molecular biology techniques to insert two random 20 nucleotide barcode sequences on the plasmid, the UP and DOWN barcodes, which juxtapose a unique *Pmi*I restriction enzyme site on the plasmid (Methods). Both the UP barcodes and DOWN barcodes contain common PCR priming sites for rapid amplification of all barcodes from a pooled sample. We generated a dual-barcoded vector library with ~250,000 clones in *E. coli* DH10B and characterized this library by associating the barcode pairs using BPseq. The vector library of ~250,000 clones was sufficient to map unique barcode pairs with confidence and also to yield a Dub-seq library in which each fragment will have a unique barcode (see below).

To generate the *E. coli* Dub-seq library, we extracted *E. coli* (BW25113) genomic DNA, sheared to 3-kb fragment size, and cloned the fragments in cloning strain DH10B into the dual-barcoded backbone vector digested with *Pmi*I. Both *E. coli* BW25113 and *E. coli* DH10B are derivatives of *E. coli* K-12. The

*E. coli* BW25113 Dub-seq library encompasses ~40,000 vectors, corresponding to about 8X coverage of the *E. coli* genome. In this study, we depend on the endogenous *E. coli* transcription and translation apparatus to drive the expression of the encoded gene (s) within each genomic fragment, although future studies could use inducible systems (for example, when the source of the cloned Dub-seq DNA differs from the host bacterium for assaying fitness[42]). The phenotypes we observe derive from increased gene copy number (that will typically result in overexpression of the genes encoded on the fragment) but other potential effects such as toxicity associated with the gene overexpression[43] or titration of DNA-binding transcription factors due to increased copy number of regulatory regions are possible[16,44]. Here, we use the term "overexpression" throughout with the caveat that increased gene dosage may not always lead to increased expression[16,44].

We next characterized the *E. coli* BW25113 Dub-seq library using BAGseq, which identifies the cloned genome fragment and its pairings with the neighboring dual barcodes. As there are two barcodes for each Dub-seq library, we performed two separate BAGseq sample preparation steps, one for the UP barcodes and one for the DOWN barcodes. Briefly, BAGseq involves shearing of the Dub-seq plasmid library, end repair, Illumina adaptor ligation, PCR amplification of the junction between the barcode and genomic insert using primers that are complementary to one of the barcode-specific primer binding sites, and deep sequencing of these samples (modified from reference [13]). After filtering out barcodes that mapped to more than one genomic fragment, we identified 30,558 unique barcode pairs that we could confidently associate with a genomic fragment.

In the *E. coli* BW25113 Dub-seq library, the fragments are largely evenly distributed across the chromosome (Fig. 2a), the

average fragment size is 2.6 kb (Fig. 2b), and the majority of fragments covered 2–3 genes in their entirety (Fig. 2c). Eighty percent of genes in the *E. coli* genome are covered (from start to stop codon) by at least five independent genomic fragments in the Dub-seq library (Fig. 2d) and 97% of all genes are covered by at least one fragment. Just 135 genes are not covered in their entirety by any Dub-seq fragment (Supplementary Data 1). Many of these unmapped or uncovered genes encode membrane and ribosomal proteins and probably reflect the lethality of overexpressing these genes[45]. Other genes could not be confidently mapped because they are associated with repetitive regions. For example, we could not confidently map fragments covering ETT2 type III secretion system pathogenicity island and its regulator gene *ygeH*, which has tetratricopeptide repeat motifs, whereas the neighboring protein-coding genes are well mapped (Fig. 2a). Similarly, we could not map genes within ribosomal RNA operons (for example, *rrlD*, Fig. 2a), as *E. coli* encodes multiple nearly identical copies of these loci. Some large genes with length >3.5 kb, such as *rpoB*, are not entirely covered by any fragments in our library, whereas other large genes such as *acrB* are covered by only one fragment (Fig. 2a).

Out of the 303 *E. coli* protein-coding genes that were shown essential for viability in previous studies[5], 95% are completely covered by at least one fragment in the Dub-seq library (Supplementary Data 2). There are only 17 protein-coding genes that are both essential for viability when deleted and absent from our Dub-seq library (Supplementary Data 2).

**Strain and gene fitness profiling using BarSeq**. The key advantage of Dub-seq is the ease of assessing the relative fitness contributions of all genes contained in the cloned genomic fragments using pooled, competitive growth assays. Depending on the assay condition and the gene(s) encoded by a genomic fragment, the relative abundance of a strain carrying that fragment can change due to its fitness advantage or disadvantage relative to strains carrying other fragments. Because the DNA barcodes have been previously associated with each genomic fragment, we can simply compare the relative abundance of each barcode before and after selective growth using DNA barcode sequencing or BarSeq[41].

As a demonstration of Dub-seq fitness assays and to illustrate our approach for calculating strain (fragment) and gene fitness scores, we recovered an aliquot of the *E. coli* BW25113 Dub-seq library cloned in *E. coli* DH10B strain in Luria-Bertani (LB) liquid medium to mid-log phase, collected a cell pellet for the "start" (or time-zero sample), and used the remaining cells to inoculate an LB culture supplemented with 1.2 mM nickel. After growth in the presence of nickel, we collected a second cell pellet for the "condition" sample. We extracted plasmid DNA from the start and condition samples, PCR amplified the UP and DOWN DNA barcodes from each, and sequenced the DNA barcodes with Illumina. We calculate the fragment fitness score for each strain by taking the normalized log2 ratio of the number of reads for each barcode in condition sample versus the start sample (Fig. 1). Positive scores indicate that the gene(s) contained on that fragment lead to an increase in relative fitness, whereas negative values mean the gene(s) on the fragment reduced relative fitness. Scores near zero indicate no fitness reduction or benefit for the gene(s) under the assayed condition (although overexpression of a gene at a different level might have an effect). As in previous work[46], we find that fitness scores calculated with either UP barcodes or DOWN barcodes yield very similar results ($r = 0.94$, Supplementary Fig. 1a, b). Therefore, we only sequenced the UP barcodes for all additional experiments in this study.

Given that multiple, causative and non-causative genes can be contained on a single fragment, to assign a fitness score to a particular gene it is necessary to examine the score of all fragments containing the gene. Here, we considered two different ways to estimate fitness score of a gene. The first approach was to simply take the average of all fitness scores for fragments that contained the gene in its entirety (the "mean" score). The second approach was to use a regression method for estimating gene fitness score so as to prevent genes from having artifactually high fitness scores if they were located near other causative genes. Specifically, we adopted non-negative least squares (NNLS) regression (the "regression" score) (see Methods). To illustrate how the mean and regression scores differ in practice, consider the gene fitness scores for two adjacent genes under elevated nickel stress, *rcnA* and *rcnR* (Fig. 3a, b). RcnA is a nickel efflux protein whose overexpression is known to lead to increased nickel tolerance[47]. Conversely, *rcnR* encodes a transcriptional repressor that weakly represses its own expression and that of *rcnA*, and the overexpression of *rcnR* alone is not expected to increase nickel tolerance[47]. Although the mean and regression approaches both result in similar (and correct) high Dub-seq scores for *rcnA* (Fig. 3a), only the regression approach results in the correct, neutral fitness score for the *rcnR* (Fig. 3b). The mean score calculation approach leads to an artifactually high fitness score for *rcnR* because many of the fragments that contain this gene also contain the neighboring *rcnA* (Fig. 3b, Supplementary Figs. 2a, b and 3a, b). Based on these results and other examples (Supplementary Fig. 4) that we examined, we concluded that the optimal strategy was to use the regression method for calculating Dub-seq gene fitness scores (Methods).

To assess the reproducibility of Dub-seq fitness assays, we compared the results obtained from independent samples. First, the number of sequencing read counts for each UP barcodes from the Dub-seq library from different start samples were highly correlated (Supplementary Fig. 1c). Similarly, between two biological replicates of the nickel stress experiment, we found a strong correlation for fragment fitness ($r = 0.80$; Fig. 3c) and for regression-based gene fitness ($r = 0.89$; Fig. 3d).

**Fitness profiling across dozens of experimental conditions**. To demonstrate the scalability of Dub-seq, we performed 155 genome-wide pooled fitness experiments representing 52 different chemicals: 23 compounds as the sole source of carbon in a defined growth media and varying concentrations of 29 inhibitory compounds in rich media (Fig. 4). The inhibitory compounds included metals, salts, and antibiotics. For each of these assays, we compared the abundance of the UP barcodes before and after growth selection. We multiplexed 48 or 96 BarSeq PCR samples per lane of Illumina sequencing, at a sequencing cost of about $20 per genome-wide assay. In the typical condition sample, we obtained ~4.2 million BarSeq reads, representing ~100 reads on an average for each clone in the Dub-seq plasmid library. We computed gene fitness scores (using the regression approach) for 4027 protein-coding genes and for 124 RNA genes. The gene fitness scores were reproducible, with a median pairwise correlation of 0.80 across 64 biological replicates.

We classified effects as high-confidence if |score| >= 2, there was sufficient read coverage, and the effects were consistent across fragments that cover the gene and/or across replicate experiments (see Methods). At a false discovery rate of 2%, we identified 4051 high-confidence effects, representing 813 of the 4151 genes assayed (Methods, Supplementary Data 3). Four hundred different genes had a high-confidence fitness benefit when overexpressed in at least one condition, whereas the overexpression of 570 different genes led to a decrease in fitness

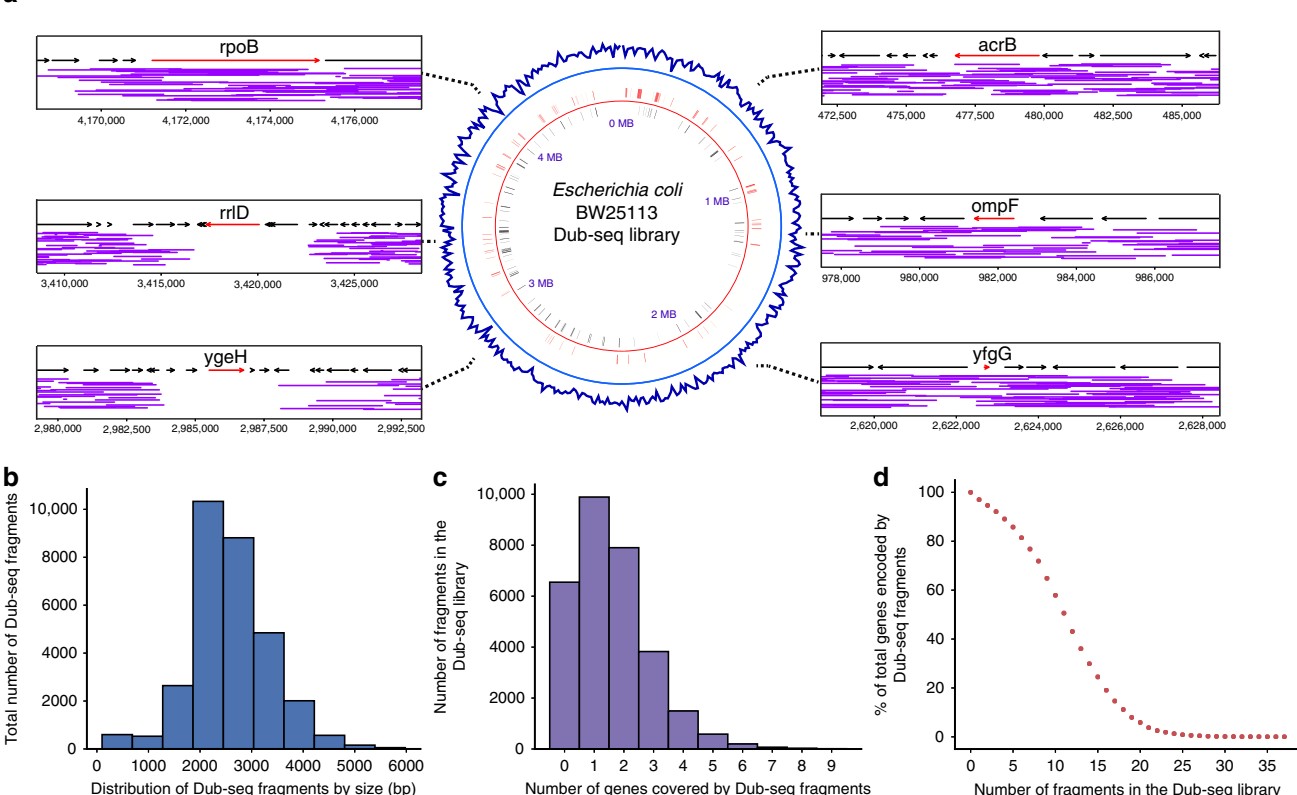

**Fig. 2** *E. coli* dual-barcoded shotgun expression library sequencing (Dub-seq) library characterization. **a** Center: genomic coverage of the *E. coli* BW25113 Dub-seq library in 10 kb windows (blue track). Black and red line-tracks represent genes essential for viability when deleted[5] that are encoded on the negative and positive strands, respectively, and are covered in the Dub-seq library. Left and right: regions of the *E. coli* chromosome covering *acrB, ompF, yfgG, ygeH, rrlD,* and *rpoB*. Each purple line represents a Dub-seq genomic fragment (the *y* axis is random). **b** The fragment insert size distribution in the *E. coli* Dub-seq library. **c** The distribution of number of genes that are completely covered (start to stop codon) per genomic fragment in the *E. coli* Dub-seq library. **d** Cumulative distribution plot showing the percentage of genes in the *E. coli* genome (*y* axis) covered by a number of independent genomic fragments (*x* axis). Source data are provided as a Source Data file

in at least one condition. Nearly all experiments (153 of 155) had at least one gene with a high-confidence effect. As the current Dub-seq design relies on the native promoters within the fragments to drive the expression of encoded genes, we looked at genes with high-confidence effect and their location within operons. Overall, genes at the beginning of transcripts were significantly more likely to have high-confidence effects than later genes in an operon (30% versus 13%, $P = 2 \times 10^{-9}$, Fisher's exact test). Nevertheless, there were 61 genes without a (known) promoter nearby that had a high-confidence benefit (Methods). Among the 303 *E. coli* protein-coding genes that were shown essential for viability in previous studies[5], 46 have a high-confidence benefit in at least in one experiment, demonstrating that gain-of-function approaches like Dub-seq can identify conditional phenotypes for genes that are not typically interrogated by loss-of-function approaches such as Tn-seq.

Some genes had positive fitness benefits across many conditions. In particular, five genes (*recA, galE, dgt, rcnA, fabB*) had high-confidence benefits in 10 or more different conditions. The most frequent benefits were found for *recA* and *galE*, which are disrupted in the DH10B derivative host strain we used[48] (Methods). Even for pleiotropic genes, we find that they confer a more extreme beneficial phenotype in some conditions. For example, UDP-glucose 4-epimerase (*galE*) is highly beneficial at high copy numbers in the presence of 0.1 mM benzethonium chloride, with gene scores of + 12 or + 14 in two replicate experiments. All of *galE*'s other scores were under + 5. Similarly,

strand exchange and recombination gene *recA* shows high fitness scores of + 6 in the presence of cisplatin, lomefloxacin and sodium chloride. In addition to these examples, we found that 32 genes provide growth advantage in five or more antibiotics, metals or other stress conditions, as compared with 241 genes showing growth benefit in just one condition (Supplementary Data 3).

Some of the Dub-seq experiments identified dozens of genes with high-confidence fitness benefits. For example, with potassium acetate as the carbon source, we identified 56 genes that had high-confidence fitness benefits in both of two replicate experiments (Supplementary Data 3). The two highest-scoring genes encode isozymes of aconitase (*acnA* and *acnB*), which are part of the tricarboxylic acid cycle for oxidizing acetate[49]. But the relationship between the other beneficial genes and acetate catabolism is not obvious. As another example, in copper (II) chloride stress at 2 mM, 120 genes had high-confidence benefits. The genes with the highest scores were *envZ, mltD, dpiA, mepM, cutC, ompX, ompC, ompF,* and lipoprotein *nlpE* (Supplementary Data 2 and 3). Overexpression of most of these genes is known to activate the complex regulatory network of envelope stress response via *cpxAR* and sigma-E[50,51]. Specifically, the increased copper tolerance of strains that overexpress *nlpE* or *cutC* are due to activation of the Cpx pathway[52] or the sigma-E response[53], respectively. Finally, dozens of genes show growth benefits in the presence of the membrane-disrupting cationic surfactants benzethonium and benzalkonium. Most of these genes are

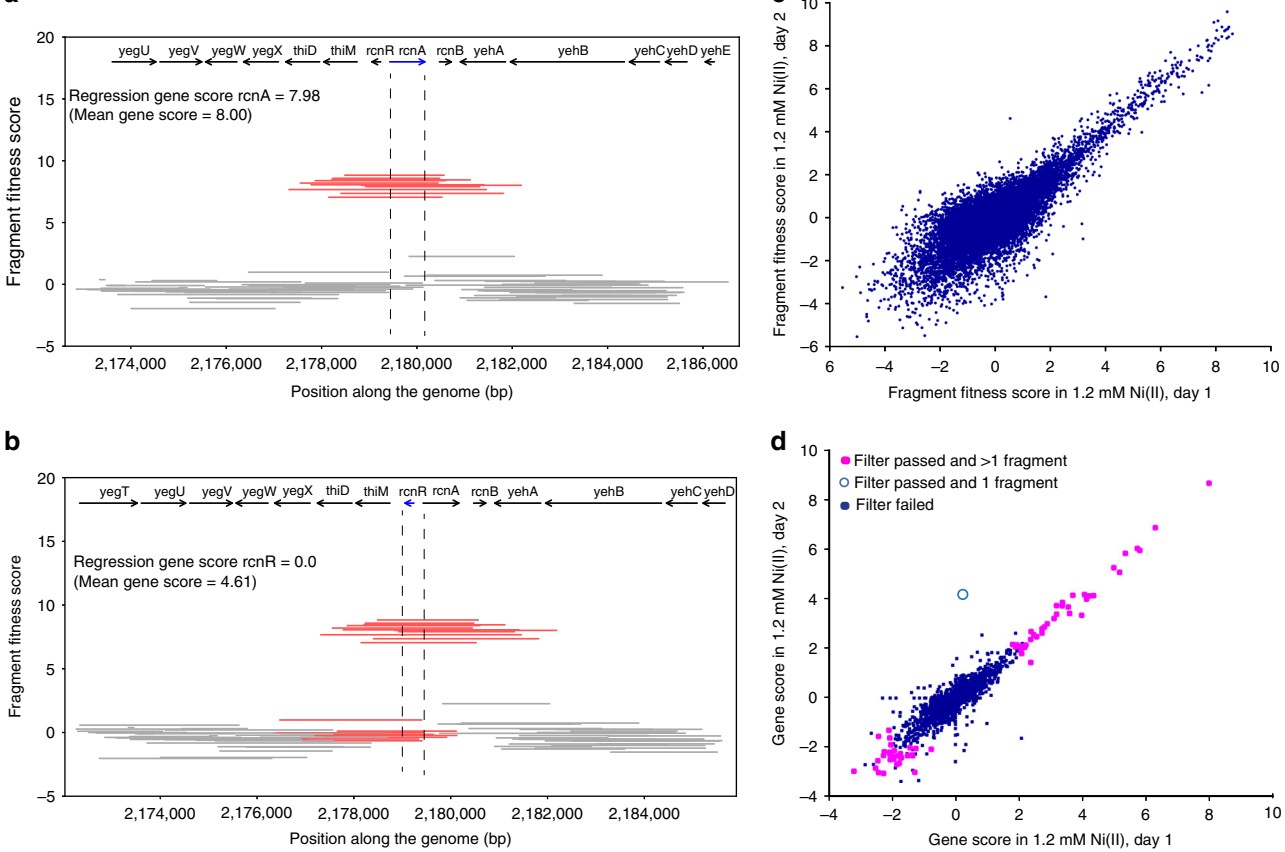

**Fig. 3** Fragment and gene fitness dual-barcoded shotgun expression library sequencing (Dub-seq) scores. **a** Dub-seq fragment (strain) data for region surrounding *rcnA* under elevated nickel stress (*y* axis). Each line shows a Dub-seq fragment. Those that completely cover *rcnA* are in red. Both the mean and regression scores reflect the known biology of *rcnA* as a nickel resistance determinant[47]. **b** Same as (**a**) for the neighboring *rcnR*, which encodes a transcriptional repressor of *rcnA*. Fragments that cover *rcnR* are in red. **c** Comparison of fragment fitness scores for two biological replicates of 1.2 mM nickel stress. **d** Same as (**c**) for gene fitness scores calculated using the regression approach. Genes are highlighted if their score met our criteria for a high-confidence effect within an experiment (without considering other experiments) (see Methods); we also show whether the gene score is based on just one fragment (effects from just one fragment are not considered high confidence if they do not replicate). Source data are provided as a Source Data file

involved in membrane lipid homeostasis, envelope stress response pathways, and drug efflux systems (Fig. 4, Supplementary Data 3).

In total, we identified 41 instances where the Dub-seq fitness data are consistent with the known growth benefit imparted by the gene (Supplementary Data 4). These high confidence, known hits include genes encoding diverse functions such as efflux pumps, transporters, and regulators, as well as biosynthetic enzymes and small RNAs, each yielding enhanced fitness via diverse mechanisms. For example, overexpression of *cysE* (which encodes serine acetyltransferase) probably increases nickel tolerance through increased glutathione biosynthesis[54], whereas overexpression of *rnc* (which encodes RNase III) yields a growth benefit in nickel and cobalt stress, as it downregulates the expression of *corA*, which encodes a transporter that mediates the influx of nickel and cobalt ions into the cell[55].

We also identified high-confidence fitness benefits for hundreds of genes that had not been previously associated with a tolerance phenotype, including *pssA*, *dcrA/sdaC*, and *dcrB* in sisomicin; *pmrD* in aluminum; *treA*, *treB*, and *phnM* in phosphomycin; sRNAs *chiX* in nickel and *ryhB* in zinc; and a number of genes of unknown function (Fig. 4, Supplementary Data 3). To follow up some of the novel observations, we assayed the growth of strains overexpressing the genes individually with and without added stress. We used *murA* overexpression as a test case, as this is known to confer resistance to phosphomycin[56] (Supplementary Fig. 5). Growth curves confirmed that the

overexpression of either *pssA* or *dcrB* confers resistance to the aminoglycoside antibiotic sisomicin, although the mechanism(s) by which this resistance is conferred remains unclear. The gene *pssA* encodes an essential phosphatidylserine synthase, whereas *dcrB* is a periplasmic protein with a role in phage infection[49]. Growth curves also confirm that the overexpression of the outer membrane protein MipA confers strong resistance to benzethonium chloride (Supplementary Fig. 5). *mipA* has previously been implicated in the resistance to other antibiotics[57].

Gene overexpression can also decrease host fitness[16,17,58] and may indicate important function for those gene products. We identified 570 genes with a high-confidence negative effect on fitness in at least one experiment (Supplementary Data 3). Some of these genes appear to be more generally toxic when overexpressed or have a global regulatory role and compromise host fitness in multiple conditions. Twenty-four genes had detrimental effects on fitness in 10 or more different conditions (Supplementary Data 3). Conversely, some genes have negative gene scores in a few conditions. For example, consistent with earlier studies we found that overexpression of *glpT* or *uhpT* increases susceptibility to phosphomycin[59]. These results also agree with clinical data, which shows that the main cause of phosphomycin resistance in patients is the downregulation of GlpT via downregulation of cAMP[59]. Accordingly, we also found that overexpression of *cpdA* (which encodes an enzyme that hydrolyzes cAMP) enhances fitness under phosphomycin stress (Fig. 4).

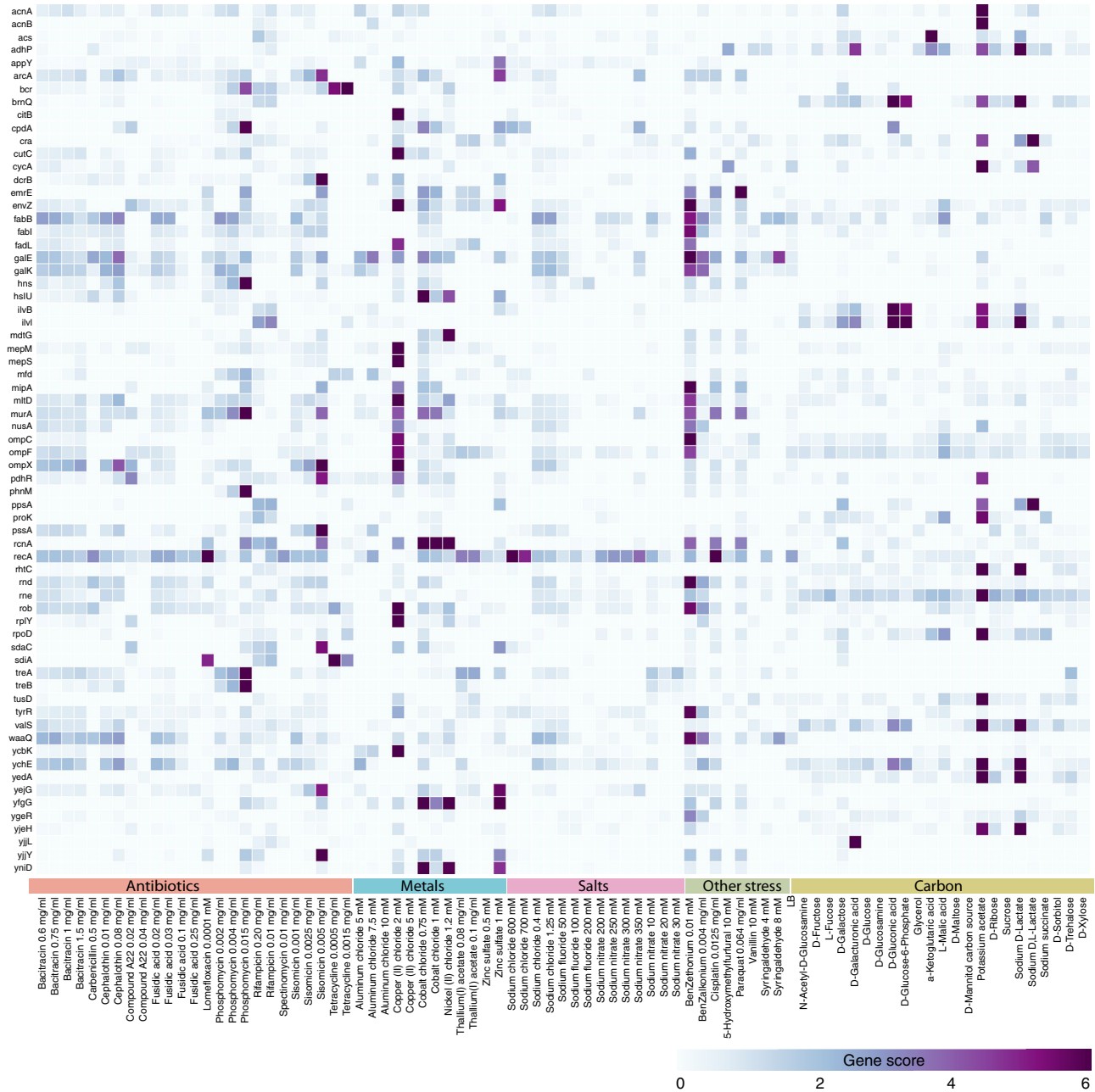

**Fig. 4** Heatmap of dual-barcoded shotgun expression library sequencing (Dub-seq) fitness data for 52 conditions and for 67 genes with large benefits. Only genes with a high-confidence effect and gene fitness score >= 6 in at least one condition are shown. Gene scores from two replicate experiments were averaged. Source data are provided as a Source Data file

Finally, we analyzed our data for "epistatic" instances where multiple nearby genes are necessary for the observed phenotype. Specifically, we searched for evidence of synergy between genes by analyzing scores for fragments containing more than one gene that are significantly greater than the inferred sum of score of the constituent genes (Methods). In total, we found six high scoring epistatic-effect cases across 52 conditions in our Dub-seq data set (*fetA-fetB* on nickel, *ampD-ampE* on benzethonium, *ackA-pta* on D-lactate, *arcA-yjjY* on sisomicin, *hns-tdk* on phosphomycin, and *yfiF-trxC* on potassium acetate (Supplementary Fig. 6a–c)). Among these, three gene pairs have related functions (*fetA-fetB* form a complex, *pta-ackA* encode enzymes that catalyze adjacent reactions in the catabolism of lactate, and *ampD-ampE* are thought to be a signaling pathway[49]) and our data indicate, together they provide a larger growth benefit. Specifically,

overexpression of *fetAB* together has been shown to improve survival during nickel stress[60].

**Comparison with loss-of-function fitness data.** Integrating large-scale genetic gain- and loss-of-function can provide added specificity to biological insights. For instance, genes with resistance phenotypes when overexpressed and sensitivity phenotypes when deleted are often specifically involved in the condition of interest, as demonstrated by studies identifying drug targets in yeast[61] or identifying small RNA regulators[62] or antibiotic resistance factors in bacteria[63]. Furthermore, genes with opposing loss- and gain-of-function phenotypes for stress compounds are more likely to be true resistance determinants as opposed to genes that have indirect effects when overexpressed[16]. For 45 of the

conditions that we profiled in this study with Dub-seq, we can systematically compare these phenotypic consequences of over-expression to loss-of-function mutations as determined by random barcode transposon site mutagenesis[15]. The two data sets studied the same growth media and compounds, but not necessarily at the same concentrations, and they used different strains of *E. coli* (DH10B or BW25113). Across these 45 conditions, we identified 625 high-confidence benefits of overexpression (or 0.3% of gene-condition pairs). Of the 625 high-confidence benefits, 480 are for genes with RB-TnSeq data, and in 62 cases (12%) that loss-of-function led to a significant disadvantage (RB-TnSeq fitness $< -1$ and $t < -4$, where $t$ is a $t$-like test statistic[13]). By chance, we would expect just 2.5% agreement, which is significantly less ($P < 10^{-15}$, chi-squared test of proportions). Overall, we found moderate overlap between genes that are beneficial when overexpressed and important for fitness when disrupted (Supplementary Data 3).

To illustrate the biological insights that can be derived by systematically comparing gain- and loss-of-function data on a genomic scale, we present three examples: growth in the presence of elevated nickel, cobalt, or sodium chloride (Fig. 5a–c). Under each condition, we find that a number of genes that are both necessary for resisting the stress when knocked-out and sufficient for a resistance phenotype when singly overexpressed. These instances include known examples such as the aforementioned metal exporter RcnA[47] and RNase III for cobalt and nickel tolerance[55], as well as the osmolyte transporter ProP[64] and envelope biogenesis factor YcbC (ElyC)[65] for tolerance to osmotic stress imposed by sodium chloride. (In our Dub-seq data, *proP* and *ycbC* failed to pass the filters for high-confidence effects). In addition to these known examples, there are more novel observations (Fig. 5a–c). Under nickel and cobalt stress, the uncharacterized protein YfgG (DUF2633) is important for tolerance, a finding that is supported by RB-Tnseq data[15] and by individual growth curve analysis of an *yfgG* overexpression strain (Fig. 5d). Although the precise biochemical function of YfgG is unclear, a close homolog of this protein in *Klebsiella michiganensis* is also important for fitness under nickel and cobalt stress[15]. As a second example, we find that ProY is important for nickel resistance. A ProY homolog in the related bacterium *K. michiganensis* is also important for nickel resistance[15]. Using individual strain growth curve analysis, we confirmed that overexpression of *proY* alone can confer nickel resistance to *E. coli* (Fig. 5e). Although ProY is currently annotated as a cryptic proline transporter, we suspect that its function is to transport histidine as it can suppress histidine auxotrophy[25] and homologs of this protein are required for histidine utilization in other bacteria[15]. In light of this, we speculate that the nickel resistance phenotype of ProY is due to increased sequestration of nickel ions by a higher intracellular concentration of histidine. As a final example, we found that the porphyrogen oxidase YfeX confers sodium chloride resistance in *E. coli*, a finding confirmed by an individual growth curve analysis (Fig. 5f). Although we are unsure how this protein manifests this phenotype, we note that yfeX homologs are important for resisting sodium chloride in multiple bacteria[15]. We have provided a general working hypothesis for many of other genes with high fitness scores in Supplementary Data 5.

## Discussion

Here we describe Dub-seq, a technology for performing parallelized gain-of-function fitness assays across diverse conditions. Dub-seq couples shotgun cloning of random DNA fragments with competitive fitness assays to assess the phenotypic importance of the genes contained on those fragments in a single tube assay. We demonstrate that Dub-seq is reproducible, economical, scalable, and identifies both known and novel gain-of-function phenotypes.

In this proof-of-concept study, we generated a Dub-seq library of *E. coli* genomic DNA and performed 152 genome-wide assays to identify 400 different genes with a high-confidence fitness benefit when overexpressed in at least one experimental condition. As far as we know, the majority of these gene–phenotype associations have not been reported before and they include dozens of genes of unknown function (Supplementary Data 3). We found 241 genes that confer a fitness benefit in just one condition, indicating a condition-specific phenotype. Thirty-two genes enhanced fitness in five or more conditions, suggesting their broader role in host fitness and importance in cross-resistance phenotypes observed between metals, antibiotics, antiseptics, and other stresses[66,67]. Dub-seq recapitulated 41 known instances of positive fitness effects, wherein the fitness phenotypes stem from diverse mechanisms (Supplementary Data 4). Finally, we show that systematically comparing gain-of-function and loss-of-function data sets provide additional insights into those genes that are both necessary and sufficient for stress tolerance phenotypes, as we illustrated for *yfgG* (a gene of unknown function important for nickel and cobalt tolerance), *proY* (a probable histidine transporter), and *yfeX* (poryphorigen oxidase important for sodium chloride tolerance). Intriguingly, all three of these examples have conserved phenotypes in other bacteria, demonstrating that even in *E. coli* there are evolutionary conserved functions that remain to be elucidated with approaches like Dub-seq.

Dub-seq can be readily extended to DNA from other sources and many cultured bacteria could be adapted as hosts for the genome-wide fitness assays. By using other hosts, we can overcome gene expression and toxicity issues associated with expressing heterologous DNA in model hosts[34–36]. To extend the Dub-seq methodology for functional profiling of metagenomic DNA isolated from diverse communities, we would need to generate and map a higher diversity of barcode pairs. In addition, to ensure reliable expression of heterologous genes, a number of approaches can be used to activate transcription or translation of genes encoded within foreign DNA[42,68].

In this work, we generated a Dub-seq library with a ~2.6 kb insert size and therefore by design, the library only covers fragments encoding 2–3 genes on an average. Almost all of the genes with Dub-seq data (98%) are covered by at least two independent fragments. As Dub-seq fragments in this work cover only 2–3 genes, the phenotypes that are only conferred by the activity of a larger group of genes (such as multisubunit complexes) will not be detected. By adapting the Dub-seq strategy to fosmids, cosmids and bacterial artificial chromosomes, future efforts can clone larger size genomic fragments to create Dub-seq libraries for the discovery of activities encoded by multiple genes, including secondary metabolites and screened in diverse model host organisms.

Given the increasing knowledge gap between genomic sequence and function, and the limited ability of computational approaches to accurately predict gene function from sequence, high-throughput experimental methods are needed to assign gene function and resolve roles of uncharacterized genes. Recently, a number of loss-of-function methods have been developed[5–8,10–14], but only a fraction of genes from genetically tractable microbes can be readily annotated with a specific function using these approaches. We envision that multiple, complementary experimental approaches that can be applied *en masse* and that corresponding improvements in computational tools are ultimately necessary to not only uncover the roles of most microbial genes, but also to propagate these new annotations into existing database structures. The Dub-seq approach we presented here fulfills

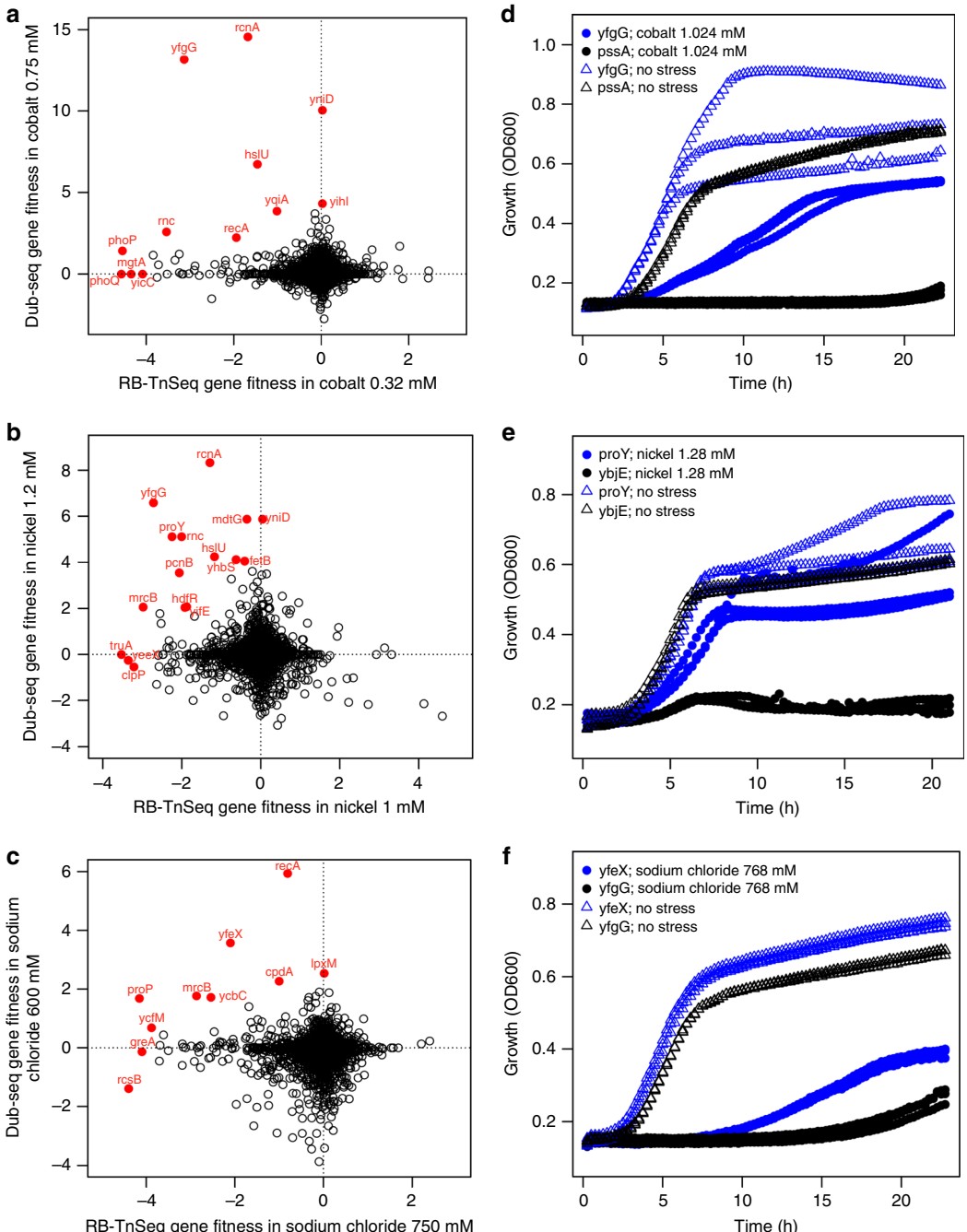

**Fig. 5** Comparing genome-wide loss- and gain-of-function phenotype data. **a–c** Comparison of RB-TnSeq fitness data[15] (x axis) and dual-barcoded shotgun expression library sequencing (Dub-seq) gene fitness data for *E. coli* genes under growth with inhibitory concentrations of cobalt (**a**), nickel (**b**), and sodium chloride (**c**). Selected genes are highlighted. **d** Growth of *E. coli* overexpressing *yfgG* under cobalt stress; *pssA* is a control. **e** Growth of *E. coli* overexpressing *proY* under nickel stress; *ybjE* is a control. **f** Growth of *E. coli* overexpressing *yfeX* under sodium chloride stress; *yfgG* is used as a control. Growth curves in (**d**), (**e**), and (**f**) are averaged from three replicate experiments. Source data are provided as a Source Data file

an important technology gap in performing gain-of-function screens and will facilitates systematic functional characterization of microbial genomes.

## Methods

**Strains and growth conditions**. *E. coli* BW25113 was purchased from the *E. coli* Genetic Stock Center. All plasmid manipulations were performed using standard molecular biology techniques[69]. All enzymes were obtained from New England Biolabs (NEB) and oligonucleotides were received from Integrated DNA Technologies (IDT). *E. coli* strain DH10B (DH10B derivative, NEB 10-Beta) was used for plasmid construction and as host for Dub-seq fitness assays. Unless noted, all strains were grown in LB supplemented with 30 μg/ml chloramphenicol at 37 °C in

the Multitron shaker. The primers, plasmids, and strains used in this study are listed in Supplementary Data 6, 7 and 8, respectively.

**Construction of dual-barcoded Dub-seq vector**. To construct a double-barcoded vector, we used pFAB5477 an in-house medium copy (copy number of ~20) plasmid with modified pBBR1 replication origin and a chloramphenicol resistance marker[70]. pBBR1-based broad host plasmids are relatively small, mobilizable and have been widely used for a variety of genetic engineering applications in diverse microbes[71]. To insert a pair of DNA barcodes on the plasmid, we used phosphorylated oFAB2853 and oFAB2854 primers to amplify the entire plasmid pFAB5477, removed the plasmid backbone using *DpnI* (as per manufacturing instructions, NEB), and ligated the amplified and pure product using T4 ligase (as

per manufacturing instructions, NEB). The random N's in oFAB2853 and oFAB2854 (Supplementary Data 6) represent the UP and DOWN barcode sequences. The ligated product, pFAB5491, was column purified using the Qiagen PCR purification kit, transformed into DH10B electrocompetent cells (NEB 10-Beta *E. coli* cells, as per manufacturing instructions, NEB) and transformants were selected on LB agar plates supplemented with 30 μg/ml chloramphenicol. The next day, ~250,000 colony forming units (CFU) were estimated and scraped together into 20 ml LB with 30 μg/ml chloramphenicol. The culture library was diluted to an optical density at 600 nm (OD600) of 0.2 in fresh LB medium supplemented with 30 μg/ml chloramphenicol and grown to a final OD600 of ~1.2. We added glycerol to a final concentration of 15%, made multiple 1 ml glycerol stocks, and stored them at –80 °C. We also collected cell pellets to prepare plasmid DNA of pFAB5491 for further characterization of the library (BPseq).

**BPseq to characterize dual-barcoded Dub-seq vector.** To associate the pair of DNA barcodes, we performed BPseq of the plasmid pFAB5491 library. For deep coverage of the library, we performed 10 different PCR reactions using primers VM_barseq_P1 and VM_Barseq-P2. The forward primers VM_Barseq-P2 contains different 6-bp TruSeq indexes, and were automatically demultiplexed by the Illumina software.

We performed PCR in a 100-μl total volume with 5 μl common reverse primer VM_barseq_P1 (4 μM), 5 μl forward primer VM_Barseq-P2 _IT001 to IT010 (4 μM), 38 μl of sterile water, 2 μl template pFAB5491, and 50 μl of 2X stock of Q5 DNA Polymerase mix (500 μl of 2X stock of Q5 DNA Polymerase mix consists of 200 μl Q5 buffer, 20 μl dNTP, 50 μl DMSO, 10 μl Q5 DNA Polymerase enzyme, and 220 μl water) under following PCR conditions: 98 °C for 4 min, followed by 15 cycles of 30 s at 98 °C, 30 s at 55 °C, 30 s at 72 °C, and final extension at 72 °C for 5 min. Finally, we ran the PCR products on an analytical gel to confirm amplification. We pooled equal volumes (10 μl) of BarSeq PCR products, purified the combined product using Qiagen PCR purification kit, and eluted in 40 μl of sterile water. We quantified the DNA product with a Qubit double-stranded DNA (dsDNA) high-sensitivity (HS) assay kit (Invitrogen). The BPseq samples were sequenced first on Illumina MiSeq and then HiSeq 2500: both with 150-bp single-end runs.

**BPseq data analysis.** BPseq reads were analyzed with *bpseq* script from the *Dub-seq* python library with default parameters (code available at https://github.com/psnovichkov/DubSeq). The script looks for the common flanking sequences around each barcode (UP and DOWN) and requires an exact match of nine nucleotides on both sides. By default, these flanking sequences may be up to two nucleotides away from their expected positions. The script also requires that each position in each barcode have a quality score of at least 20 (that is, an estimated error rate of under 1%). This gives an initial list of pairs of barcodes with the correct length and reliable sequence quality.

We applied two additional filters to minimize the number of erroneous barcode pairs that can be caused by PCR artifacts or sequencing errors. First, we check whether a given barcode can be a result of a single-nucleotide substitution introduced in a real barcode and filter out all such barcodes. We perform a pairwise sequence comparison of all extracted barcodes (UP and DOWN barcodes are treated separately) and search for "similar" barcodes. Two barcodes are considered to be *similar* if they are different by only one nucleotide. A given barcode passes the filter if it does not have similar barcodes or it is at least two times more frequent than the most abundant similar barcode.

Second, we check whether a given barcode pair can be a result of chimeric PCR and filter out all such pairs. As the region between and around UP and DOWN barcodes are identical in all plasmids in our library, we expected artifacts from formation of chimeric BPseq PCR products[13]. We perform a pairwise comparison of all barcode pairs and search for "related" pairs. Two-barcode pairs are considered to be related if they have either the same UP or DOWN barcodes. The presence of the same UP (or DOWN) barcode in multiple barcode pairs is potentially a sign of chimeric PCR. To distinguish the true barcode pair from the chimeric one, we check the frequency of all the related barcode pairs. A given barcode pair passes the filter and is considered to be non-chimeric if it does not have related pairs or it is at least two times more frequent than the most abundant related barcode pair. As a result, the "reference set" of barcode pairs is created. From the BPseq step, we obtained 5,436,798 total reads. Among these, total usable reads (reads that support barcode pairs from the reference set) were 2,933,702 and represent about 54% of total reads.

**Dub-seq vector preparation for cloning genomic fragments.** To prepare the Dub-seq vector pFAB5491 for cloning, we made 900 μl or about 100 μg of plasmid preparation (Qiagen plasmid miniprep kit), and performed two rounds of *Pmi*I digestion. Restriction digestion reaction included 900 μl (total 100 μg) of pFAB5491 plasmid, 100 μl *Pmi*I enzyme, 400 μl 10X cutsmart buffer, and water to make up the volume of 4000 μl. We incubated the reaction at 37 °C on a heating block for 4 h and then checked the reaction progress on an analytical 1% agarose gel. To dephosphorylate the restriction-digested vector, we added one unit of rSAP for every 1 pmol of DNA ends (about 1 μg of a 3-kb plasmid), and incubated at 37 °C for 2 h in a PCR machine. We stopped the reaction by heat inactivation of rSAP

and restriction enzyme at 70 °C for 20 min. The cut and dephosphorylated vector library was then gel purified (Qiagen gel extraction kit). To remove any uncut vector, we repeated the entire process of restriction digestion, dephosphorylation, and purification. The final concentration of cut and pure barcoded vector library used for cloning genome fragments was about ~30 ng/μl.

**Construction of *E. coli* Dub-seq library.** To construct Dub-seq library of *E. coli* genomic fragments, we extracted *E. coli* BW25113 genomic DNA and 1 μg was fragmented by ultrasonication to an average size of 3000 bp using a Covaris S220 focused ultrasonicator. The sheared genomic DNA was then gel purified to size select and end-repaired using End-IT kit (Epicentre, as per manufacturer instruction). Briefly the 50 μl reaction included: 34 μl sheared DNA (1.0 μg total), 5 μl ATP 10 mM, 5 μl dNTP mix (10 mM), 5 μl EndIt buffer 10X, and 1–2 μl EndIT enzyme. We incubated the reaction at room temperature for 45 min, and inactivated the enzyme by incubating the reaction at 70 °C for 10 min. The end-repaired genome fragments were purified with PCR clean-up kit (Qiagen), and quantified on Nanodrop.

The end-repaired genomic fragments were then ligated to the restriction-digested, sequence-characterized dual-barcoded backbone vector (pFAB5491) at 8:1 insert:vector ratio using Fast-link Ligase enzyme (Epicentre, as per manufacturer instruction). The total 60 μl ligation reaction consists of 4 μl of restriction-digested pFAB5491, 20 μl end-repaired DNA, 3 μl ATP (10 mM), 6 μl 10X ligase buffer, 19 μl water, and 8 μl Fast-link-ligase. The ligation was incubated overnight (18 h) at 16 °C, inactivated at 75 °C for 15 min, and purified using PCR purification kit (Qiagen).

For transforming the ligation reaction, 60 μl of column-purified ligation reaction was mixed gently with 1500 μl of NEB DH10B electrocompetent cells on ice and then the mix was dispensed 60 μl per cuvette. Electroporation was done using parameters supplied by NEB. Transformed cells were recovered by adding 1 ml super optimal recovery media (as per competent cell manufacturer instruction, NEB). We pooled all recoveries and added additional 10 ml of fresh SOC. Transformants were then incubated at 37 °C with shaking for 90 min. We spun down the pellets and resuspended the pellet in 6 ml SOC. Different volumes of 6 ml resuspended pellets were then plated on overnight-dried bioassay plates (Thermo Scientific # 240835) of LB agar supplemented with 30 μg/ml chloramphenicol. We also did dilution series for estimating CFUs.

We determined the number of colonies required for 99% coverage of *E. coli* genome using the formula $N = \ln(1-0.99)/\ln(1-(\text{Insert size/Genome size}))$ to ensure that genome fragments are present in the cloned library[72]. For example, to cover the *E. coli* genome (of size 4.7 Mb) with fragments of 3 kb, we need about 4610 strains for 99% coverage. We collected ~40,000 colonies by scraping the colonies using a sterile spatula into 20 ml LB supplemented with 30 μg/ml chloramphenicol in a 50 ml Falcon tube and mixed well. This *E. coli* Dub-seq library was then diluted to an optical density at 600 nm (OD600) of 0.2 in fresh LB supplemented with 30 μg/ml chloramphenicol and grown to a final OD600 of ~1.2 at 37 °C. We added glycerol to a final concentration of 15%, made multiple stocks of 1 ml volume, and stored the aliquots at –80 °C. We also made cell pellets to store at –80 °C and to make large plasmid preparation (Qiagen) for BAGseq library preparation.

**BAGseq to characterize barcoded genomic fragment junctions.** We characterized the final plasmid library pFAB5516 using a TnSeq-like protocol[13], which we call Barcode-Association-with Genome fragment sequencing or BAGseq. BAGseq identifies the cloned genome fragment and its pairings with neighboring dual barcodes. This step of associating the dual barcodes with each library of genomic fragments is only done once (by deep sequencing) and used as a reference table to derive connections between observed functional/fitness traits with specific cloned genome fragment (Fig. 1).

To generate Illumina-compatible sequencing libraries to link both UP and DOWN random DNA barcodes to the ends of the cloned genome fragments, we processed two samples per library. The plasmid library (1 μg) samples were fragmented by ultrasonication to an average size of 300 bp with a Covaris S220 focused ultrasonicator. To remove DNA fragments of unwanted size, we performed a double size selection using AMPure XP beads (Beckman Coulter) according to the manufacturer's instructions. The final fragmented and size-selected plasmid DNA was quality assessed with a DNA-1000 chip on an Agilent Bioanalyzer. Illumina library preparation involves a cascade of enzymatic reactions, each followed by a cleanup step with AMPure XP beads. Fragmentation generates plasmid DNA library with a mixture of blunt ends and 5′ and 3′ overhangs. End repair, A-tailing, and adapter ligation reactions were performed on the fragmented DNA using the NEBNext DNA Library preparation kit for Illumina (New England Biolabs), according to the manufacturer's recommended protocols. For the adapter ligation, we used 0.5 μl of a 15 μM double-stranded Y adapter, prepared by annealing Mod2_TS_Univ (ACGCTCTTCCGATC*T) and Mod2_TruSeq (Phos-GATCGGAAGAGCACACGTCTGAACTCCAGTCA). In the preceding oligonucleotides, the asterisk and Phos represent phosphorothioate and 5′ phosphate modifications, respectively. To specifically amplify UP barcodes and neighboring genomic fragment terminus by PCR, we used the UP-tag-specific primer oFAB2923_Nspacer_barseq_universal, and P7_MOD_TS_index1 primer. For the DOWN-tag amplification, we used oFAB2924_ Nspacer_barseq_universal

and P7_MOD_TS_index2 primer. For the BAGseq UP barcode and DOWN barcode site enriching PCR, we used JumpStart Taq DNA polymerase (Sigma) in a 100 μl total volume with the following PCR program: 94 °C for 2 min and 25 cycles of 94 °C 30 s, 65 °C for 20 s, and 72 °C for 30 s, followed by a final extension at 72 °C for 10 min. The final PCR product was purified using AMPure XP beads according to the manufacturer's instructions, eluted in 25 μl of water, and quantified on an Agilent Bioanalyzer with a DNA-1000 chip. Each BAGseq library was then sequenced on the HiSeq 2500 system (Illumina) with a 150 SE run to map UP and DOWN barcodes to genomic inserts in the Dub-seq E. coli library.

**BAGseq data analysis**. BAGSeq reads were analyzed with *bagseq* script from the *Dub-seq* python library with default parameters (code available at https://github.com/psnovichkov/DubSeq). Fastq files for UP and DOWN barcodes with associated (cloned) genomic fragments are processed separately. For each read, the script looks for the flanking sequences around a barcode and requires an exact match of nine nucleotides on both sides and a minimum quality score of 20 for each nucleotide in a barcode. The sequence downstream of the identified barcode is considered to be a candidate genomic fragment and is required to be at least 15 nucleotides long for further processing. As a result, the initial list of the extracted barcodes and candidate genomic fragments is constructed.

All extracted genomic fragments were compared with the E. coli genome sequence with BLAT using default parameters. Only hits with alignment block size of at least 15 nucleotides and at most one indel were considered. It is also required that the extracted genomic fragment is mapped to one location in the genome. Thus, mappings to repeat regions were ignored. We applied two additional filters to minimize the number of erroneous associations between barcode and genomic location. First, we applied the same type of filter that we use for the analysis of BPSeq reads to filter out barcodes with a one-nucleotide error.

Second, the same barcode can be associated with different genomic fragments because of PCR artifacts (chimeras) or because multiple fragments were cloned between the same pair of barcodes. To filter out erroneous barcode mappings, the number of reads supporting different locations for the same barcode were calculated. To distinguish the true location from the false one, the frequency of the most abundant location (the number of supported reads) was compared with frequencies of all other locations for the same barcode. A given association between the barcode and the genomic location is considered to be true if the barcode does not have any other associated locations or the abundance of this association is at least two times more frequent than any other associations for the same barcode. As a result, the reference set of associations between UP (and separately for DOWN) barcodes and genomic locations is created, which we call "BAGseq reference set".

The BPseq reference set of barcode pairs and BAGseq reference set are combined together to associate pairs of barcodes with genomic regions (to create the final "Dub-seq reference set"). This step is done using the *bpag* script from the *Dub-seq* python library with default parameters. For each BPseq barcode pair, the script checks if the associations between UP and DOWN barcodes with genomic locations are present in the BAGSeq reference set. If both UP and DOWN barcodes (from BPseq reference set) are mapped to the genome, then the script checks the length of the region between the mapped locations and requires it to be between 100 nt and 6 kb. As a result, the final Dub-seq reference list of barcode pairs associated with genomic regions is created. Among total 10,600,088 reads for UP barcodes, usable reads were 3,884,931 (BAGseq UP barcode reads supporting the Dub-seq reference set), representing about 36.65% of total reads, whereas for total 9,671,635 reads for DOWN barcodes, usable reads were 2,499,399, representing about 25.84% of total reads (BAGseq DOWN barcode reads supporting the Dub-seq reference set).

**Competitive growth experiments**. For genome-wide competitive growth experiments, a single aliquot of the Dub-seq library in E. coli DH10B was thawed, inoculated into 25 ml of LB medium supplemented with chloramphenicol (30 μg/ml), and grown to mid-log phase. At mid-log phase, we collected cell pellets as a common reference for BarSeq (termed start or time-zero samples) and we used the remaining cells to set up competitive fitness assays under different experimental conditions at a starting OD600 of 0.02. For carbon source growth experiments, we used M9 defined medium supplemented with 0.3 mM ʟ-leucine (as DH10B is auxotrophic for ʟ-leucine)[48] and chloramphenicol. For experiments with stress compounds, we used an inhibitory but sublethal concentration of each compound, as determined previously[15]. All stress experiments were done in LB with chloramphenicol. All pooled fitness experiments were performed in 24-well microplates with 1.2 ml of media per well and grown in a multitron shaker. We took OD readings periodically in a Tecan M1000 instrument to ensure that the cells were growing and to confirm growth inhibition for the stress experiments. The assayed Dub-seq library cell pellets were stored at –80 °C prior to plasmid DNA extraction.

**BarSeq**. Plasmid DNA from Dub-seq library samples was extracted either individually using the Plasmid miniprep kit (Qiagen) or in 96-well format with a QIAprep 96 Turbo miniprep kit (Qiagen). Plasmid DNA was quantified with the Quant-iT dsDNA BR assay kit (Invitrogen). The BarSeq PCR of UP barcodes was done as previously described[13] with ~50 ng of plasmid template per BarSeq PCR reaction. To quantify the reproducibility of both UP and DOWN barcodes in competitive growth experiments, we collected plasmid DNA from nickel and cobalt

experiments, and amplified both UP and DOWN barcodes in two separate PCRs using the same plasmid library template. For BarSeq PCR of DOWN barcodes, we used universal-forward-primer DT_BarSeq_p1_FW and reverse primer DT_Bar-Seq_IT017. The PCR cycling conditions and purification steps were same as for the UP barcodes[13]. All experiments done on the same day and sequenced on the same lane are considered as a "set".

**BarSeq data analysis and fragment score calculation**. From HiSeq 4000 runs, we obtained ~400 million of reads per lane, or 4.2 million reads per sample (for multiplexing 96 samples) typically >60% reads were informative after filtering out reads for sequencing errors and unmapped barcodes. BarSeq reads were analyzed with *barseq* script from the *Dub-seq* python library with default parameters. For each read, the script looks for the flanking sequences around each barcode and requires an exact match of nine nucleotides on both sides and a minimum quality score of 20 for each nucleotide in a barcode. The number of reads supporting each barcode is calculated. We apply the same type of filter that we use for the analysis of BPSeq reads to filter out barcodes with single-nucleotide substitutions relative to real barcodes (see BPSeq section). As a result, the list of barcode and their counts is created.

**Calculation of fragment scores (fScores)**. Given a reference list of barcodes mapped to the genomic regions (BPSeq and BAGSeq), and their counts in each sample (BarSeq), we estimate fitness values of each genomic fragment (strain) using *fscore* script from the Dub-seq python library with default parameters. First, the script identifies a subset of barcodes mapped to the genomic regions that are well represented in the time-zero samples for a given experiment set. We require that a barcode have at least 10 reads in at least one time-zero sample to be considered a valid barcode for a given experiment set. Then, the *fscore* script calculates fitness score only for the strains with valid barcodes.

Strain fitness ($f_i$) is calculated as a normalized $\log_2$ ratio of counts between the treatment (condition or end) sample $s_i$ and sum of counts across all (start) time-zero $t_i$

$$f_i = \log_2\left(\frac{s_i + 1}{t_i + 1}\right)$$

Then, the strain fitness scores are normalized so that the median in each experiment is zero.

**Calculating gene score (gScore)**. Given the fitness scores calculated for all Dub-seq fragments, we estimate a fitness score for each individual gene that is covered by at least one fragment. As mentioned in the Results, simply averaging the scores for the fragments that cover a gene gives spurious results for non-causative genes that are adjacent to a causative gene. To overcome this problem, we modeled the fitness score of each fragment as the sum of the fitness scores of the genes that are completely covered by this fragment. Our model for estimating gene scores assumes that genes contribute independently to fitness, that most genes have little impact on fitness, and that intergenic regions have no effect on host fitness.

To estimate gene scores, we cannot use ordinary least squares (OLSs), the most common type of regression, because of over fitting, which would produce unrealistic high positive and low negative scores for many genes. We also considered regularization methods (Ridge, LASSO, and ElasticNet), but these suffered from either too much shrinkage of fitness scores (biasing them towards zero) or failed to eliminate over fitting (see Supplementary Note 1, Supplementary Fig. 7). Instead, we use NNLS regression[73], where the predicted gene scores are restricted to take only non-negative values. If a gene with a potential benefit is next to (but not covered by) a fragment with negative fitness, most regression methods would inflate the benefit of the gene and assign a negative score to the nearby gene. NNLS instead ignores the (often noisy) negative scores for the nearby fragments. To estimate negative gene scores, we also used NNLS, but with the signs of the fragment scores flipped.

In our model, the expected fitness of a fragment is given by

$$f_i = \sum_j g_{ij}$$

were $g_{ij}$ is a fitness score of a gene covered by $i$-th fragment completely. The NNLS minimizes

$$||Ag - f||_2^2, \text{ subject to } g \geq 0$$

where $g$ a vector of gene fitness scores to be estimated, $f$ is vector of the "observed" fitness scores of fragments, $A$ a matrix of ones and zeros defining which gene is covered by which fragment completely. Gene scores were calculated using the *gscore* script from the Dub-seq python library with default parameters, which uses the nnls function from the *optimize* package of the *scipy* python library.

**High-confidence gene scores and estimating the false discovery rate**. We used several filters to identify gene scores that were likely to be of high-confidence and reliable. Whereas the non-negative regression was used to determine if the high

fitness of the fragments covering the gene are due to this gene or a nearby gene, these filters were intended to ensure that the fragments covering the gene had a genuine benefit. Briefly, we identify a subset of the effects to be reliable, if the fitness effect was large relative to the variation between start samples ($|score| >= 2$); the fragments containing the gene showed consistent fitness (using a $t$-test); and the number of reads for those fragments was sufficient for the gene score to have little noise (see below). Effects that passed these filters were more likely to be consistent in replicate experiments (for example, see Fig. 3d). We considered an effect that passed these filters to be of high confidence if it was based on more than one fragment or if the gene had a large effect in another experiment for the compound. In the following paragraphs, we detail these data filtration steps.

The first filter was $|gene\ score| >= 2$, as such a large effect occurred just four times in 17 control comparisons between independently processed but identical "start" samples (0.2 per experiment). In contrast, the actual conditions gave 40 large effects per experiment on average (over 150 times more).

Second, we noticed that some genes had high scores because of a single fragment with a very high score. These fragments did not have high scores in replicate experiments, so their high scores might be due to secondary mutations. To filter out these cases, we performed a single-sample $t$-test on the fragment scores (for the fragments that covered the gene) and required $P < 0.05$ (two-tailed $t$-test). This test asks if the mean is significantly different from a reference value. To handle uncertainty in the true centering of the fragment scores (which were normalized to have a median of zero), we considered the mean of all fragment scores for the experiment. We used this as the reference value (instead of zero) if this mean had the same sign as the gene's score. This makes the filter slightly more stringent. If the gene has just one fragment, then we cannot apply the $t$-test, so we instead require that $|fragment\ score|$ be in the top 1% for this experiment.

Third, we checked that the effect was larger relative to the expected noise in the mean of the fragment scores that cover the gene. The expected noise for each fragment can be estimated as $\mathrm{sqrt}(1/(1 + \mathrm{count\_after}) + 1/(1 + \mathrm{count\_start}))/\ln(2)$. This approximation is derived from the best case that the noise in the counts follows a Poisson distribution. The expected noise for the mean of the fragment scores is then $\mathrm{sqrt}(\mathrm{sum}(\mathrm{fragment\_noise}^2))/\mathrm{nfragments}$. Note that $z = \mathrm{mean}(\mathrm{fragment\ score})/\mathrm{noise}$ would (ideally) follow the standard normal distribution. We use $|z| >= 4$ as a filter; with 4303 genes being assayed, we would expect about 0.3 false positives per experiment.

Effects that passed these three filters were usually consistent across replicate experiments and represent reliable scores. We had two biological replicates for 64 of the 82 conditions (a compound at a given concentration) that we studied. Across these 64 pairs of replicate experiments, 85% of genes with filtered effects in one replicate were consistent ($|score| >= 1.5$ and the same sign) in the other replicate. Large effects ($|score| >= 2$) were more likely to replicate if they were filtered (85% versus 59% otherwise). Among filtered effects for genes covered by more than one fragment, 39% of the effects that did not replicate were from a single condition (zinc sulfate stress at 1 mM). We did not identify any obvious issue for the data from this condition. In total, 4303 genes are covered by at least one fragment, but there are only 4151 genes with at least one gene score (adequate representation in at least one start sample).

Effects that passed these three filters were considered to be high confidence if the gene was covered by multiple fragments. Because of the risk of secondary mutations, a measurement for a gene with a single fragment was only considered high confidence if it was reliable and was also supported by a large effect ($|score| >= 2$) in another experiment for that compound.

To estimate the false discovery rate for high-confidence effects, we randomly shuffled the mapping of barcodes to fragments, recomputed the mean scores for each gene in each experiment, and identified high-confidence effects as for the genuine data. This shuffling test will probably overestimate the false discovery rate because it assumes that all of the variability in the fragment scores is due to noise. Also, we used the mean score, rather than regression-based gene score, in this test. This might also lead to an overestimate of the false discovery rate . We repeated the shuffle procedure 10 times. On average, each shuffled data set had 75 high-confidence effects, whereas the actual data had 4051 high-confidence effects, so we estimated the false discovery rate as $75/4051 = 1.9\%$.

**Impact of operon structure on gene fitness**. In the current Dub-seq design, the plasmid backbone lacks a promoter or ribosome binding site  to drive the expression of genes within the random fragments, so expression relies on the native promoters within the fragments. If a fragment contains a gene but not its promoter, then the gene might not be expressed and might not show a benefit. In particular, genes that are in operons, but are not at the beginning of the operon, might not show a benefit. On the other hand, internal promoters within operons are common[74,75]. To determine if the lack of a promoter is a problem in practice, we asked how often genes at the beginning of transcripts or later in transcripts[76] had high-confidence fitness benefits.

To quantify the effect on gene fitness due to gene location within an operon, we made a list of genes that are found first in a transcript or later in a transcript based on the operon structures from RegulonDB 10.5 version[76]. We ignored operons with "weak" evidence confidence level. Genes that were at the beginning of one transcript and at a later position in another transcript were excluded. This filtering narrowed down the list of genes in operons to 881 that have Dub-seq data and gene

fitness scores. We compared the fitness of the first and later genes (in operons) to examine the impact of operon structure in the Dub-seq data.

**Calculating gene-pair fitness score**. Although our model assumes that the genes on a fragment contribute independently to fitness, there are cases where multiple nearby genes work together to confer a phenotype. For estimating such "epistatic" synergistic fitness contribution by neighboring pair of genes, we included additional variables in our fitness calculation to account for the contribution of pairs of adjacent genes (and their intergenic regions). For a gene-pair to qualify to be valid hit, the score for the gene-pair has to be more than the individual gene scores from single-gene regression model, scores should be consistent across replicates and should be supported by more than one fragment. After manual filtering, we found six high scoring epistatic-effect instances where gene-pairs positively contribute to the host fitness under specific condition (Supplementary Data 5). Among these, three gene-pairs have related functions (*fetA-fetB* on nickel, *ampD-ampE* on benzethonium, *ackA-pta* on D-lactate[49]) and make biological sense. However, in the other three high scoring gene-pairs *arcA-yjjY*, *hns-tdk* and *yfiF-trxC*, each gene is divergently transcribed and the reason behind combined fitness phenotype is not obvious. We speculate, the fitness phenotype in these cases may be function of intergenic regions in addition to the encoded genes.

**Experimental validation of single genes**. To experimentally validate some of top hits in our Dub-seq results, we used the ASKA ORF collection[29]. The ASKA library consists of *E. coli* ORFs cloned on a pMB1 replication origin plasmid and driven by an Isopropyl β-D-1-thiogalactopyranoside (IPTG)-inducible promoter. We extracted individual ASKA ORF plasmids from the collection, sequence confirmed and transformed the plasmids into our assay strain *E. coli* DH10B. As the plasmid copy number and the strength of promoter and ribosome binding site used in the ASKA ORF collection is different from the broad host pBBR1 plasmid system used in *E coli* Dub-seq library, we screened for an optimum IPTG levels to induce the expression of specific gene in order to study the host fitness. We grew the individual strains in 96-well microplates with 150 μl total volume per well. These plates were grown at 30 °C with shaking in a Tecan microplate reader (either Sunrise or Infinite F200) with optical density readings every 15 min.

**Library visualization tools**. We used the Dub-seq viewer tool from the *Dub-seq* python library (https://github.com/psnovichkov/DubSeq) to generate regions of the *E. coli* chromosome covering fragments (landscape mode) presented in Fig. 2a. To generate fitness score plots as shown in Fig. 3a, b, and Supplementary Figs. 4, 6 and 7, we used gene-browser mode. We used Circa software (OmGenomics) to generate genome coverage plot shown in Fig. 2a.

**Reporting Summary**. Further information on experimental design is available in the Nature Research Reporting Summary linked to this article.

**Code availability**. Code for processing and analyzing Dub-seq data is available at https://github.com/psnovichkov/DubSeq

## Data availability
Sequencing data have been uploaded to the Sequence Read Archive under Bio-Project accession number PRJNA512427 [http://www.ncbi.nlm.nih.gov/bioproject/512427]. Complete data from all experiments (read counts per barcode, fragment scores, and gene scores) are deposited here: [https://doi.org/10.6084/m9.figshare.6752753.v1]. The source data underlying Figs. 2b–d, 3c–d, 4 and 5a–f and Supplementary Figs. 1a–c, 2a, b, 3a, b, 5a–d and 6a–c are provided as a Source Data file. Link to website with supplementary information and bulk data downloads: http://genomics.lbl.gov/supplemental/dubseq18/. All other data available from the authors upon reasonable request.

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

## Acknowledgements

We thank Mahek Modi and Aaron Gupta for assisting in the initial stage of this project. The initial concepts for this project were developed by Biodesign project supported by the Office of Science (BER), U.S. Department of Energy, DE-SCOOO8812. The implementation was funded by ENIGMA, a Scientific Focus Area Program at Lawrence Berkeley National Laboratory, supported by the U.S. Department of Energy, Office of Science, Office of Biological and Environmental Research under contract DE-AC02-05CH11231.

## Author contributions

V.K.M., A.M.D. and A.P.A. conceived the project. V.K.M., A.M.D. and A.P.A. supervised the project. V.K.M. led the experimental work and managed the entire project. P.S.N. led the computational work. V.K.M., A.M.D., T.K.O., M.C. and S.C. collected data. V.K.M., P.S.N., M.N.P. and A.M.D. analyzed the fitness data. M.N.P. and A.P.A. provided advice on data processing and modeling. V.K.M., P.S.N., M.N.P., A.M.D. and A.P.A. wrote the paper.

## Additional information

**Competing interests:** V.K.M., P.S.N., A.M.D. and A.P.A. are holders of a patent on the Dub-seq technology. Patent information is as follows: Patent applicant: The Regents of the University of California, Oakland, US. Named inventor(s): Vivek K. Mutalik, Adam P. Arkin, Adam M. Deutschbauer, Pavel S. Novichkov. Application number: 15/665,226. Status: Pub. No.: US 2018/0030435 Al; Pub. Date: 1 February 2018. The remaining authors declare no competing interests.

