## [Peer Review File · Nature Communications]

Reviewers' comments:

Reviewer #1 (Remarks to the Author):

In this study, Mutalik et al developed and describe Dub-seq, which in essence a high-throughput gain-of-function approach (gene overexpression), followed by phenotyping under a broad set of conditions through fitness growth assays (assigning gene fitness scores), for functional characterization of genes. Dub-Seq uses a random shotgun library-based approach, where random pieces of DNA are cloned into a host and barcoded, making this a high throughput assay at low cost. As a test case for Dub-Seq, the authors use E. coli, by generating a Dub-seq overexpression library with total E. coli genomic DNA, followed by >150 fitness assays (52 experimental conditions), which led them to show gain-of-function phenotypes for >800 genes. The authors fitness data results (i) validated existing biological insights and (ii) revealed new findings.

In an era of homology creep and sequencing of divergent bacterial and archaeal clades yielding many genes that are uncharacterized, this approach is an important contribution to the scientific community and the fields of microbial genomics and ecology. While gain-of-function approaches and gene fitness assays in general will not unlock the function of all genes/proteins without or with poor functional predictions, high-throughput approaches applying them will no doubt provide strides towards filling and refining some of the functional annotation knowledge gaps.

As such, this is an approach with general appeal and broad utility and thus suitable for the readership of Nature Communications. The authors have experience with the development of such methods as shows by previous publications (i.e. the RB-TnSeq approach). The manuscript is for the most part written clearly and concisely with well-detailed methods. The methods look sound.

General comments:

Introduction:

The authors nicely lead into the topic of the grand challenge of functional prediction, but I cannot help but wonder why there is no mention of MAGs and SAGs, which further exacerbate this challenge. Hundreds to more recently thousands of genomes from uncultivated taxa come out in a single study and most often there is a focus on divergent clades, which makes functional predictions even more difficult. A strength of this approach is that the gene-encoding host does not have to be cultivated and as such this would be suitable for uncultivated clades. I think this could at least briefly be brought up in the introduction to emphasize that the approach does not require genetic tractability of the organisms of interest encoding the to be characterized genes etc.

Discussion:

In the discussion, there are a few things that should be worked out better and addressed. (i) The manuscript points out several nice examples, but the message of what the authors actually learned from your data could be worked out more clearly. (ii) A bit in line with this: What will one do with the fitness data and how will it be integrated in the existing knowledgebase? For example, the authors discuss genes involved in membrane lipid homeostasis, envelope stress response, drug export that show growth benefits when grown with membrane-disrupting cationic surfactants. This is great information, but what will you do with it? How can other researches easily access these data (beyond going through your manuscript and supplementary tables) looking for their favorite gene(s) of interest to see its fitness score when overexpressed under certain conditions? A discussion on how such data is being used moving forward is warranted. How can fitness score data be integration with genomic, transcriptomics data etc. How can corrections of annotations be performed (i.e.

GenBank)? What if the fitness assay results are in conflict with existing annotations? These are non-trivial issues moving forward and should at least be briefly discussed. (iii) A brief discussion on specifically the limitations use of environmental sequences and / or use of divergent taxa in *E. coli* host should be brought up. The authors, in the discussion, do touch on the expansion to environmental sequences with higher diversity of barcoded vectors and expansion to other hosts, which I think is great and a real opportunity of the approach (as mentioned earlier this should even be pointed out in the introduction), but a brief discussion on where the limitations might lay is warranted.

MINOR:

Line 73: "Advances in DNA sequencing have had a tremendous impact on microbial genomics, as thousands of genomes have now been sequenced¹". This reference, though only a few years old, seems already outdated, as we now have sequenced an order of magnitude more, so "tens of thousands" of microbial genomes (not thousands). Suggest correction and look for more recent reference. The latest IMG/M paper in NAR just came out and may be a good fit:
<https://academic.oup.com/nar/advance-article/doi/10.1093/nar/gky901/5115825>

Line 126: "if the overexpression library is being assayed in many conditions." Should this read "under" many conditions?

Line 192: "~250,000 clones in *E. coli*". Which strain of *E. coli*? Please specify (throughout the manuscript)

Figure 2: center and line 218: "the fragments are evenly distributed across the chromosome": there is at least one region that is not covered (well or at all) by the library as visible by the dip on the chromosome track in position 10-11 o'clock. Why is this region not covered or minimally covered? There are another couple of sites with low coverage, which may warrant some brief discussion/explanation.

Ok, lateron you talk about some unmapped/ uncovered regions. Maybe change the above wording to "the fragments are largely evenly distributed across the chromosome".

Figure 2B: a size selection step prior to cloning would probably tighten up the size distribution if that was desired.

Figure 2C: should the x-axis label be slightly shifted? I.e. first bar representing 1 gene being covered by the Dub-Seq fragments, with ~6000 of such 1-gene fragments being covered in the Dub-seq library? As is, it looks confusing.

Line 232: "3.5 Kb" generally kilo is lowercase (as you also spell earlier). Correct and be consistent.

Line 234: it might be worthwhile calculating a threshold for the minimum gene coverage needed for this assay to work for the given gene and then providing some stats on the % of genes that were "missed" in this assay due to no or too low coverage. I would suspect this % will be rather low and absolutely acceptable, but some level of quantification would be useful for the reader and for setting expectations and having a baseline for researchers who want to apply this to their organisms/ sequences of interest.

Line 236: "Of the E. coli protein-coding genes that are essential for viability when deleted". Can you add here how many these are? "Out of the X E. coli protein-coding genes.. ". This would be useful for the reader.

"Of the E. coli protein-coding genes that are essential for viability when deleted". This sentence sounds a bit odd. I think I came across it earlier as well. Can you reword to something like: "Of the E. coli protein-coding genes that were shown essential for viability in previous deletion studies (5)"

Figure 3 a, b: x-axis label and units missing: position along the genome (bp) or alike.

Line 302: "52 different chemicals". Figure 4 figure legend mentions 53 conditions. Is it 52 or 53? Please clarify.

Figure 4: the heatmap would be easier to interpret by the reader is if it was clustered in some biologically meaningful way along the y-axis rather than just listing the genes in alphabetical order. Can genes be arranged by at least some of their predicted functions, operon structure or alike. Some naturally are bc of their names, but many are not.

Reviewer #2 (Remarks to the Author):

The authors describe a convenient method for rapidly creating and screening strains with increased copy number of genes under a large number of growth conditions. The big attraction of the method is the ease and rapidity with which it can be done. The authors were able to identify several genes that produced phenotypes in higher copy number several of which were either validated or make biological sense.

In general, the writing is fairly clear and the outline of the methodology would allow others to do similar work. However, the analysis is rather limited and we think that the claims could be more appropriately circumscribed.

Major issues:

- This is discussed as an overexpression method but actually it's a method for increasing gene copy number. In many cases, this will result in overexpression. However, particularly when protein levels are either regulated post-transcriptionally or through a feedback mechanism on transcription itself, there might not be any overexpression. In general, that means that negative results are not meaningful. Moreover, even positive results could arise from other mechanisms. For example, increased copy number of certain elements could end up titrating out DNA binding proteins.

- The huge caveat to the analysis is that, while the authors mention operons, they do not really take transcriptional machinery and organization into account in their analysis as far as we can tell. Certainly truncated genes may or may not be active. However, as they say that their plasmid lacks a promoter and RBS, any shearing that separates a gene from its promoter will result in lack of expression. Moreover, the downstream genes of large operons should never be expressed in their system as their fragment size is too small to capture the associated genes. The analysis should take this into account. And, in fact, this also provides a quality control - these genes should virtually never be associated with a phenotype.

Minor points:

- The copy number of the plasmid should be mentioned. The literature seems to suggest that it is either low- or medium-copy.
- The authors include dual bar codes but it appears that only one is necessary for their analysis.
- Much of the discussion of "gene interactions" is overstated. The actual examples are of multisubunit complexes arranged in operons. Gene interactions that are not operonic would never be picked up by this approach.
- The authors need to present the data in a manner which follows their commentary within the text. They mention multiple observations, which are not shown and require a good deal of work by the reader to understand: 'high-confidence benefits in >10 conditions', 'putatively beneficial genes', 'known-benefit genes', 'not been previously associated with tolerance phenotype'.
- Supplementary figure 6 could be part of figure 4.
- Panels a, b and c should have a more descriptive legend for the X axis

Author's replies to reviewer comments (in Blue):

Reviewer #1:

In this study, Mutalik et al developed and describe Dub-seq, which in essence a high-throughput gain-of-function approach (gene overexpression), followed by phenotyping under a broad set of conditions through fitness growth assays (assigning gene fitness scores), for functional characterization of genes. Dub-Seq uses a random shotgun library-based approach, where random pieces of DNA are cloned into a host and barcoded, making this a high throughput assay at low cost. As a test case for Dub-Seq, the authors use E. coli, by generating a Dub-seq overexpression library with total E. coli genomic DNA, followed by >150 fitness assays (52 experimental conditions), which led them to show gain-of-function phenotypes for >800 genes. The authors fitness data results (i) validated existing biological insights and (ii) revealed new findings.

In an era of homology creep and sequencing of divergent bacterial and archaeal clades yielding many genes that are uncharacterized, this approach is an important contribution to the scientific community and the fields of microbial genomics and ecology. While gain-of-function approaches and gene fitness assays in general will not unlock the function of all genes/proteins without or with poor functional predictions, high-throughput approaches applying them will no doubt provide strides towards filling and refining some of the functional annotation knowledge gaps.

As such, this is an approach with general appeal and broad utility and thus suitable for the readership of Nature Communications. The authors have experience with the development of such methods as shows by previous publications (i.e. the RB-TnSeq approach). The manuscript is for the most part written clearly and concisely with well-detailed methods. The methods look sound.

We thank the reviewer for their kind words and comments on our work.

General comments:

Introduction:

The authors nicely lead into the topic of the grand challenge of functional prediction, but I cannot help but wonder why there is no mention of MAGs and SAGs, which further exacerbate this challenge. Hundreds to more recently thousands of genomes from uncultivated taxa come out in a single study and most often there is a focus on divergent clades, which makes functional predictions even more difficult. A strength of this approach is that the gene-encoding host does not have to be cultivated and as such this would be suitable for uncultivated clades. I think this could at least briefly be brought up in the introduction to emphasize that the approach does not require genetic tractability of the organisms of interest encoding the to be characterized genes etc.

Thanks for pointing out the applicability of Dub-seq as an experimental platform to interrogate gene function from SAGs and MAGs. Given their recalcitrance to cultivation, nearly all gene-function prediction in SAGs/MAGs is currently done computationally.

Dub-seq can be easily extended to DNA prepared from SAGs and MAGs and will help fill the gene function knowledge gap in bacteria, as we describe in the Discussion. Based on the comment of the reviewer, we now briefly mentioned these points in the revised introduction as below.

Line 143: “Given that only DNA and a suitable host organism for assaying fitness are necessary and not the genetic tractability of the organisms of interest, Dub-seq can be readily extended to diverse functional genomics and biotechnology applications including functional interrogation of DNA from uncultivated clades”

Line 544: “In particular, our dual-barcoded vectors should be suitable to build Dub-seq libraries of microbial isolates and single-amplified genomes, and can be mobilized to new bacteria via conjugation because of their broad-host range replication origin”

Line 555: “If barcoded vector libraries can be scaled to more strains and strategies for heterologous expression of foreign DNA is achieved, then Dub-seq should be extendable to metagenome-derived DNA.”

Discussion:

In the discussion, there are a few things that should be worked out better and addressed. (i) The manuscript points out several nice examples, but the message of what the authors actually learned from your data could be worked out more clearly.

We have revised the discussion of the manuscript to more clearly describe the novel contributions of this work. We have added these additional lines to the text:

Line 396: “265 genes of unknown function (**Fig. 4, Supplementary Table 3**)”

Line 536: “as we illustrated for *yfgG* (a gene of unknown function important for nickel and cobalt tolerance), *proY* (a probable histidine transporter), and *yfeX* (poryphorin oxidase gene important for sodium chloride tolerance). Intriguingly, all three of these examples have conserved phenotypes in other bacteria, demonstrating that even in *E. coli* there are evolutionary conserved functions that remain to be elucidated with approaches like Dub-seq”

Line 522: “As far as we know, the majority of these gene-phenotype associations have not previously been reported including 265 genes of unknown function (**Supplementary Table 3**)”

Line 555: “If barcoded vector libraries can be scaled to more strains and strategies for heterologous expression of foreign DNA is achieved, then Dub-seq should be readily extendable to metagenome-derived DNA.”

Line 561: “....almost all of the genes with Dub-seq data (98%) are covered by at least two independent fragments. We recommend any future applications of the approach to aim

for the similar gene coverage. As Dub-seq fragments in this work cover only 2-3 genes,....”

Line 580: “We envision that multiple, complementary experimental approaches that can be applied *en masse* and that corresponding improvements in computational tools are ultimately necessary to not only uncover the roles of most microbial genes, but also to propagate these new annotations into existing database structures.”

Line 583: “The Dub-seq approach we presented here fulfills an important technology gap in performing gain-of-function screens and will facilitate systematic functional characterization of microbial genomes.”

(ii) A bit in line with this: What will one do with the fitness data and how will it be integrated in the existing knowledgebase? For example, the authors discuss genes involved in membrane lipid homeostasis, envelope stress response, drug export that show growth benefits when grown with membrane-disrupting cationic surfactants. This is great information, but what will you do with it? How can other researchers easily access these data (beyond going through your manuscript and supplementary tables) looking for their favorite gene(s) of interest to see its fitness score when overexpressed under certain conditions? A discussion on how such data is being used moving forward is warranted. How can fitness score data be integration with genomic, transcriptomics data etc. How can corrections of annotations be performed (i.e. GenBank)? What if the fitness assay results are in conflict with existing annotations? These are non-trivial issues moving forward and should at least be briefly discussed.

We agree with the reviewer that data accessibility is very important for other researchers to access and analyze large-scale datasets. Therefore we have provided both raw and analyzed data in supplementary information (also, <http://genomics.lbl.gov/supplemental/dubseq18/>) and deposited complete data from all of the experiments presented in this work (read counts per barcode, fragment scores and gene scores) in Figshare (<https://doi.org/10.6084/m9.figshare.6752753.v1>).

The reviewer raises an important issue regarding the integration of new data and newly-derived functional annotations into existing knowledgebases. This is a big challenge for genomics and the annotation/biocuration community, as it is not currently possible to systematically turn phenotype data into annotations that are propagated to existing resources such as GenBank or UniProt. For Genbank/NCBI, edits of gene annotations are nominally restricted to the sequence depositor, not a third party generating new functional data for those genes (https://www.ncbi.nlm.nih.gov/books/NBK53704/#gbankquickstart.although_i_m_not_li sted).

From our recent work on identifying phenotypes for 11,779 protein-coding genes using loss-of-function genetics (PMID: 29769716), we were unable to add our phenotype-derived gene annotations to established databases, including UniProt, even though we presented these data to them in a simple tab-delimited format.

To highlight some of these challenges, we have included this text in the discussion to address the need for computational tools, line (580):

“We envision that multiple, complementary experimental approaches that can be applied *en masse* and that corresponding improvements in computational tools are ultimately necessary to not only uncover the roles of most microbial genes, but also to propagate these new annotations into existing database structures.”

(iii) A brief discussion on specifically the limitations use of environmental sequences and / or use of divergent taxa in E. coli host should be brought up. The authors, in the discussion, do touch on the expansion to environmental sequences with higher diversity of barcoded vectors and expansion to other hosts, which I think is great and a real opportunity of the approach (as mentioned earlier this should even be pointed out in the introduction), but a brief discussion on where the limitations might lay is warranted.

The challenges associated with heterologously expressing environmental DNA in *E. coli* have been reviewed extensively and we cite those reviews (references 34-36, 42). To emphasize this point, we added the following sentence to line 555 in the Discussion:

“If barcoded vector libraries can be scaled to more strains and strategies for heterologous expression of foreign DNA is achieved, then Dub-seq should be readily extendable to metagenome-derived DNA”

MINOR:

Line 73: “Advances in DNA sequencing have had a tremendous impact on microbial genomics, as thousands of genomes have now been sequenced¹”. This reference, though only a few years old, seems already outdated, as we now have sequenced an order of magnitude more, so “tens of thousands” of microbial genomes (not thousands). Suggest correction and look for more recent reference. The latest IMG/M paper in NAR just came out and may be a good fit: <https://academic.oup.com/nar/advance-article/doi/10.1093/nar/gky901/5115825>

Based on the reviewer’s helpful recommendations, we updated the reference and we changed “thousands” to “tens of thousands”.

Line 126: “if the overexpression library is being assayed in many conditions.” Should this read “under” many conditions?

The reviewer is correct and we made the recommended edit to the manuscript. The line 127 reads now as: “Unfortunately, sequencing the cloned regions (to identify the genes conferring the phenotype) is labor intensive and may become cost-prohibitive if the overexpression library is being assayed under many conditions”.

Line 192: “~250,000 clones in *E. coli*”. Which strain of *E. coli*? Please specify (throughout the manuscript)

In this revised version of the manuscript, we mention that the genomic DNA we cloned in the overexpression vector is from *E. coli* strain BW25113, while we cloned this library and performed genome-wide fitness assays in the DH10B *E. coli* strain. To clarify this explicitly, we also added this line (199) in the Results section “Both *E. coli* BW25113 and *E. coli* DH10B are derivatives of *E. coli* K-12.”

Figure 2: center and line 218: “the fragments are evenly distributed across the chromosome”: there is at least one region that is not covered (well or at all) by the library as visible by the dip on the chromosome track in position 10-11 o clock. Why is this region not covered or minimally covered? There are another couple of sites with low coverage, which may warrant some brief discussion/ explanation. Ok, lateron you talk about some unmapped/ uncovered regions. Maybe change the above wording to “the fragments are largely evenly distributed across the chromosome”.

Based on the reviewer’s suggestion, we changed the text in question. Line 226 reads now as “In the *E. coli* BW25113 Dub-seq library, the fragments are largely evenly distributed across the chromosome.....”

Figure 2B: a size selection step prior to cloning would probably tighten up the size distribution if that was desired.

In this work, we did select the fragment size while shearing the genomic DNA (Covaris) and also during the extraction of sheared DNA through gel preparation. We modified the text in the Methods section to clarify this point: (line 792) as “The sheared genomic DNA was then gel purified to size select and end-repaired using End-IT kit”

Figure 2C: should the x-axis label be slightly shifted? I.e. first bar representing 1 gene being covered by the Dub-Seq fragments, with ~6000 of such 1-gene fragments being covered in the Dub-seq library? As is, it looks confusing.

Fig 2C is a distribution plot with each bin representing 1 gene. We agree with the reviewer that our previous labeling of the x-axis was confusing. We have moved the x-axis labels to just under each bar to make this figure more clear.

Line 232: “3.5 Kb” generally kilo is lowercase (as you also spell earlier). Correct and be consistent.

We modified the text as suggested by the reviewer.

Line 234: it might be worthwhile calculating a threshold for the minimum gene coverage needed for this assay to work for the given gene and then providing some stats on the % of genes that were “missed” in this assay due to no or too low coverage. I would suspect

this % will be rather low and absolutely acceptable, but some level of quantification would be useful for the reader and for setting expectations and having a baseline for researchers who want to apply this to their organisms/ sequences of interest.

From our experience with Dub-seq, it is much easier to believe the data when there is >1 strain covering the gene. So, we think that at least 2 independent strains representing each gene should be the "minimum gene coverage" to aim for. To clarify this point we have added the following sentence to the Discussion at line 561: "...and almost all of the genes with Dub-seq data (98%) are covered by at least two independent fragments. We recommend any future applications of the approach to aim for the similar gene coverage."

Line 236: "Of the E. coli protein-coding genes that are essential for viability when deleted". Can you add here how many these are? "Out of the X E. coli protein-coding genes.. ". This would be useful for the reader. "Of the E. coli protein-coding genes that are essential for viability when deleted". This sentence sounds a bit odd. I think I came across it earlier as well. Can you reword to something like: "Of the E. coli protein-coding genes that were shown essential for viability in previous deletion studies (5)"

We have modified the text (line 244) as suggested by the reviewer: "Out of the 303 E. coli protein-coding genes that were shown essential for viability in previous studies"

Figure 3 a, b: x-axis label and units missing: position along the genome (bp) or alike.

We have included x-axis label and units "position along the genome (bp)" in Figs 2, 3, Supplementary Figs 4, 6, and 7. We thank the reviewer for pointing out the omission.

Line 302: "52 different chemicals". Figure 4 figure legend mentions 53 conditions. Is it 52 or 53? Please clarify.

The library was assayed in 52 chemicals and one no stress control, that is LB rich media (total 53 conditions). We edited line in the Figure 4 legend for clarity (line 666).

Figure 4: the heatmap would be easier to interpret by the reader is if it was clustered in some biologically meaningful way along the y-axis rather than just listing the genes in alphabetical order. Can genes be arranged by at least some of their predicted functions, operon structure or alike. Some naturally are bc of theirs names, but many are not.

We explored both options: ordering genes by functional category and by chromosome position (including operons). For the functional categorization, it was challenging to place genes of unknown function or genes with multiple functions into simple categories. For chromosome position, few genes in the heatmap are in the same operon as another gene, so we felt that this was not very useful. We found that listing the genes in alphabetical order would enable the reader to rapidly examine the genes we present and discuss in the text.

Reviewer #2 (Remarks to the Author):

The authors describe a convenient method for rapidly creating and screening strains with increased copy number of genes under a large number of growth conditions. The big attraction of the method is the ease and rapidity with which it can be done. The authors were able to identify several genes that produced phenotypes in higher copy number several of which were either validated or make biological sense.

In general, the writing is fairly clear and the outline of the methodology would allow others to do similar work. However, the analysis is rather limited and we think that the claims could be more appropriately circumscribed.

We thank the reviewer for their comments and feedback.

Major issues:

- This is discussed as an overexpression method but actually it's a method for increasing gene copy number. In many cases, this will result in overexpression. However, particularly when protein levels are either regulated post-transcriptionally or through a feedback mechanism on transcription itself, there might not be any overexpression. In general, that means that negative results are not meaningful. Moreover, even positive results could arise from other mechanisms. For example, increased copy number of certain elements could end up titrating out DNA binding proteins.

We have clarified the text to emphasize that Dub-seq is a shotgun library on a plasmid with medium copy number, and that the phenotypes we are seeing might be due to increased copy number of the genes and not due to overexpression per se.

We have added these lines to the text (Line 206): “The phenotypes we observe derive from increased gene copy number (that will typically result in overexpression of the genes encoded on the fragment) but other potential effects such as toxicity associated with the gene overexpression⁴³ or titration of DNA-binding transcription factors due to increased copy number of regulatory regions are possible^{16,44}. Here, we use the term ‘overexpression’ throughout with the caveat that increased gene dosage may not always lead to increased expression^{16,44}”

We agree with the reviewer that a fitness value near 0 (i.e. no phenotype) in the Dub-Seq data is not a guarantee that the overexpression of the gene will have a fitness benefit or defect under a different context (for example, inducible overexpression of a single ORF). To emphasize this, we added the following sentence to the revised manuscript line 272: “Scores near zero indicate no fitness reduction or benefit for the gene(s) under the assayed condition (although overexpression of a gene at a different level might have an effect).”

- The huge caveat to the analysis is that, while the authors mention operons, they do not really take transcriptional machinery and organization into account in their analysis as

far as we can tell. Certainly truncated genes may or may not be active. However, as they say that their plasmid lacks a promoter and RBS, any shearing that separates a gene from its promoter will result in lack of expression. Moreover, the downstream genes of large operons should never be expressed in their system as their fragment size is too small to capture the associated genes. The analysis should take this into account. And, in fact, this also provides a quality control - these genes should virtually never be associated with a phenotype.

Even though we agree with the reviewer that ‘any shearing that separates a gene from its promoter will result in lack of expression’, their assertion that ‘the downstream genes of large operons should never be expressed in their system’ is based on the assumption that there are no overlapping or intergenic promoters or cryptic “promoter-like” motifs in the cloned random DNA fragments. Recent studies have highlighted the disparate promoter-like regions throughout *E coli* genome and that they may have different sigma-factor binding efficiencies when cloned into a new genomic context (for example, PMIDs:14529615; 17096598).

To specifically address reviewer comments, we analysed our Dub-seq fitness dataset for genes within operons. We identified 386 first and 495 later genes (within operons) that have Dub-seq data. The first genes in operons were significantly more likely to have high-confidence effects than later genes (30% vs. 13%, $P = 2e-9$, Fisher exact test). So, even though the presence of a promoter is a significant issue, we have plenty of cases of genes that are known to be inside operons, yet we find phenotypes using Dub-seq. As we note in the Discussion, future iterations of Dub-seq could include inducible promoters to drive expression.

To clarify these points, we have added these lines in the Results section (Line 424):
“In the current Dub-seq design, the plasmid backbone lacks a promoter or RBS to drive the expression of genes within the random fragments, so expression relies on the native promoters within the fragments. If a fragment contains a gene but not its promoter, then the gene might not be expressed and might not show a benefit. In particular, genes that are in operons, but are not at the beginning of the operon, might not show a benefit. On the other hand, internal promoters within operons are common^{60,61}. To determine if the lack of a promoter is a problem in practice, we asked how often genes at the beginning of transcripts or later in transcripts⁶² had high-confidence fitness benefits. Genes at the beginning of transcripts were significantly more likely to have high-confidence effects than later genes (30% vs. 13%, $P = 2e-9$, Fisher exact test). Nevertheless, there were 61 genes without a (known) promoter nearby that had a high-confidence benefit.”

We also included this analysis in the Methods section, line 1076:

“Impact of operon structure on gene fitness

To quantify the effect on gene fitness due to gene location within an operon, we made a list of genes that are found first in a transcript or later in a transcript based on the operon structures from RegulonDB 10.5 version⁶². We ignored operons with "weak" evidence confidence level. Genes that were at the beginning of one transcript and at a later position

in another transcript were excluded. This filtering narrowed down the list of genes in operons to 881 that have Dub-seq data and gene fitness scores. We compared the fitness of the first and later genes (in operons) to examine the impact of operon structure in the Dub-seq data.”

Minor points:

- The copy number of the plasmid should be mentioned. The literature seems to suggest that it is either low- or medium-copy.

Based on the comment of the reviewer we now mention the copy number of the plasmid in the methods section as:

“To construct a double barcoded vector, we used pFAB5477, an in-house medium copy (copy number of ~20) plasmid with a modified pBBR1 replication origin and a chloramphenicol resistance marker⁶⁹”

- The authors include dual bar codes but it appears that only one is necessary for their analysis.

The reviewer is correct, we used the two barcodes to precisely map the junctions between cloned region and plasmid backbone, but we only need one barcode for fitness assays via Barseq analysis. So, future extensions of Dub-seq could use just a single barcode, although the mapping of the vector library (to associate the DNA barcode to the entire cloned fragment) would likely need to utilize long-read DNA sequencing technology like PacBio or Oxford Nanopore.

- Much of the discussion of "gene interactions" is overstated. The actual examples are of multisubunit complexes arranged in operons. Gene interactions that are not operonic would never be picked up by this approach.

The reviewer is correct that “gene interactions” cannot be detected for pairs of genes that are not physically located together in the chromosome with the current Dub-seq design. To clarify this point we have edited the text in the revised manuscript (line 436 as: “Finally, we analyzed our data for ‘epistatic’ instances where multiple nearby genes are necessary for the observed phenotype”

- The authors need to present the data in a manner which follows their commentary within the text. They mention multiple observations, which are not shown and require a good deal of work by the reader to understand: ‘high-confidence benefits in >10 conditions’, ‘putatively beneficial genes’, ‘known-benefit genes’, ‘not been previously associated with tolerance phenotype’.

To simplify, we have made sure to link the data (specific example, fitness number and refer the appropriate Supplementary data/Table) with commentary in the text.

We have made following changes to this revised manuscript:

Line 364: “....56 genes that had high-confidence benefits in both of two replicate experiments..)” replaced with “....56 genes that had high-confidence fitness benefits in both of two replicate experiments..)”

Line 348: “....five genes (*recA*, *galE*, *dgt*, *rcnA*, *fabB*) had high-confidence benefits in 10 or more different conditions” replaced with “five genes (*recA*, *galE*, *dgt*, *rcnA*, *fabB*) had high-confidence fitness benefits in 10 or more different conditions”

Line 369: "putatively beneficial genes" replaced with "genes with high-confidence benefits”

- *Supplementary figure 6 could be part of figure 4.*

We are worried that combining these two figures could be confusing to the reader. Supplementary figure 6 shows gene-pair fitness scores, while Figure 4 shows single gene fitness scores. For this reason, we have kept these figures separate.

- *Panels a, b and c should have a more descriptive legend for the X axis*

Reviewer 1 also noted this omission and we have now included x-axis label and units “position along the genome (bp)” in Figs 2, 3, Supplementary Figs 4, 6, and 7 and their panels. Thanks for pointing out the omission.

REVIEWERS' COMMENTS:

Reviewer #1 (Remarks to the Author):

The authors have addressed my concerns satisfactory. This study should be published with Nature Comm.

Reviewer #2 (Remarks to the Author):

The authors have responded well to the critiques. We still have one minor quibble about operon structure. While we agree that there might be cryptic promoters (which shows the limitations of using RegulonDB), the concern still stands. And it's particularly marked for protein complexes that are encoded in large operons where such cryptic promoters might be less likely to exist (e.g., ribosomal protein operons, RNA polymerase, ATP synthase). Nevertheless, this is not a indictment of the method, simply a limitation.

Author's reply to reviewers' comments:

Reviewer #1 (Remarks to the Author):

The authors have addressed my concerns satisfactory. This study should be published with Nature Comm.

We sincerely thank the reviewers for their critical feedback.

Reviewer #2 (Remarks to the Author):

The authors have responded well to the critiques. We still have one minor quibble about operon structure. While we agree that there might be cryptic promoters (which shows the limitations of using RegulonDB), the concern still stands. And it's particularly marked for protein complexes that are encoded in large operons where such cryptic promoters might be less likely to exist (e.g., ribosomal protein operons, RNA polymerase, ATP synthase). Nevertheless, this is not a indictment of the method, simply a limitation.

Our description of Dub-seq method's limitation in the discussion section already addresses this concern raised by the reviewer. Specifically, on line 684 we mention ".....the phenotypes that are only conferred by the activity of a larger group of genes (such as multisubunit complexes) will not be detected"